# Sustained interfacial powering through self-generated mantle and siphon of a gelling droplet

Chunmei Zhou[1,2], Caihong Liu[1,2], Rui Shi[3], Hongxuan Liang[1,2], Hongtu Tan[1,2], Kai Zhuang[1,2], Jiakun Guo[1,2] & Xin Tang [1,2] ✉

Autonomous motion in a persistent manner such as spinning of Euler's disk is long-sought-after by natural or artificial microsystems due to their limited energy loading and is particularly challenging for Marangoni motors as inhomogeneity of active molecules is difficult to sustain. Here we show that by releasing a droplet containing hydrogel precursor and non-small active molecules on a diluted crosslinking-agent solution, the droplet self-propels with a lifetime 300-to-1000-fold longer. It is found that continuously cross-linking hydrogel shell cuts rapid surfactant diffusion and accompanying volumetric contraction perforates the shell and generates a vent through which active molecules are unidirectionally released. The mechanism echoes squid's jet propulsion wherein water is expelled out of a siphon by contracting mantle. Such self-generated contracting mantle-siphon configuration of a gelling droplet maximizes the localized concentration inhomogeneity and protracts adsorption saturation on water surface, improving the efficiency and lifetime of Marangoni motors for sustained powering of interfacial machines. The unfolded strategy potentially provides solutions for microscale release control which will be of interest to microrobots, materials assembly, and biomolecules transport.

Persistent motion in an autonomous manner, such as spinning of Euler's disk, is long-sought-after in small-scale robots because of their limited energy loading. For the terrain of physiological or engineered air-liquid interface, the Marangoni motor is intensively studied due to its simplicity and inherent compatibility with biphasic interface[1–4]. Such locomotion is fueled by short-lived surfactant non-uniformity, a nonequilibrium state which will be rapidly homogenized by advection and diffusion. Without assistance from specially designed chemistry, the Marangoni motor is plagued by a short lifetime of ~10s, limiting its practical application as a modular power source[1,5]. It is found that squid harnesses delicate jet propulsion for locomotive agility and endurance (Fig. 1a). To propel, water is first sucked into its muscular mantle cavity. Then the squid

contracts its mantle and expels water out of the narrow siphon[6]. Utilizing such a contracting-mantle-siphon configuration, the squid, *Todarodes pacificus*, can continuously swim for months at an average speed of ~1 body length (BL) per second[7,8].

Inspired by squid's propelling skill, we unfold a general strategy to prolong the Marangoni motor's lifetime by 300-to-1000-fold using an artificial contracting-mantle-siphon system that is self-generated by a gelling droplet. By releasing a 10-μl droplet containing hydrogel precursor and non-small active molecules (low-molecular-weight amphiphilic polymer) on a diluted crosslinking-agent solution, we observe that the denser droplet floats and propels at an average velocity of ~10 mm s⁻¹ for ~100 min, a lifetime ~600 fold longer than that of conventional Marangoni motors such as camphor boats (Fig. 1b, c,

[1]Centre for Complex Flows and Soft Matter Research, Southern University of Science and Technology, Shenzhen, Guangdong, China. [2]Department of Mechanics and Aerospace Engineering, Southern University of Science and Technology, Shenzhen, Guangdong, China. [3]College of Professional and Continuing Education, The Hong Kong Polytechnic University, Hong Kong, China. ✉e-mail: tangx@sustech.edu.cn

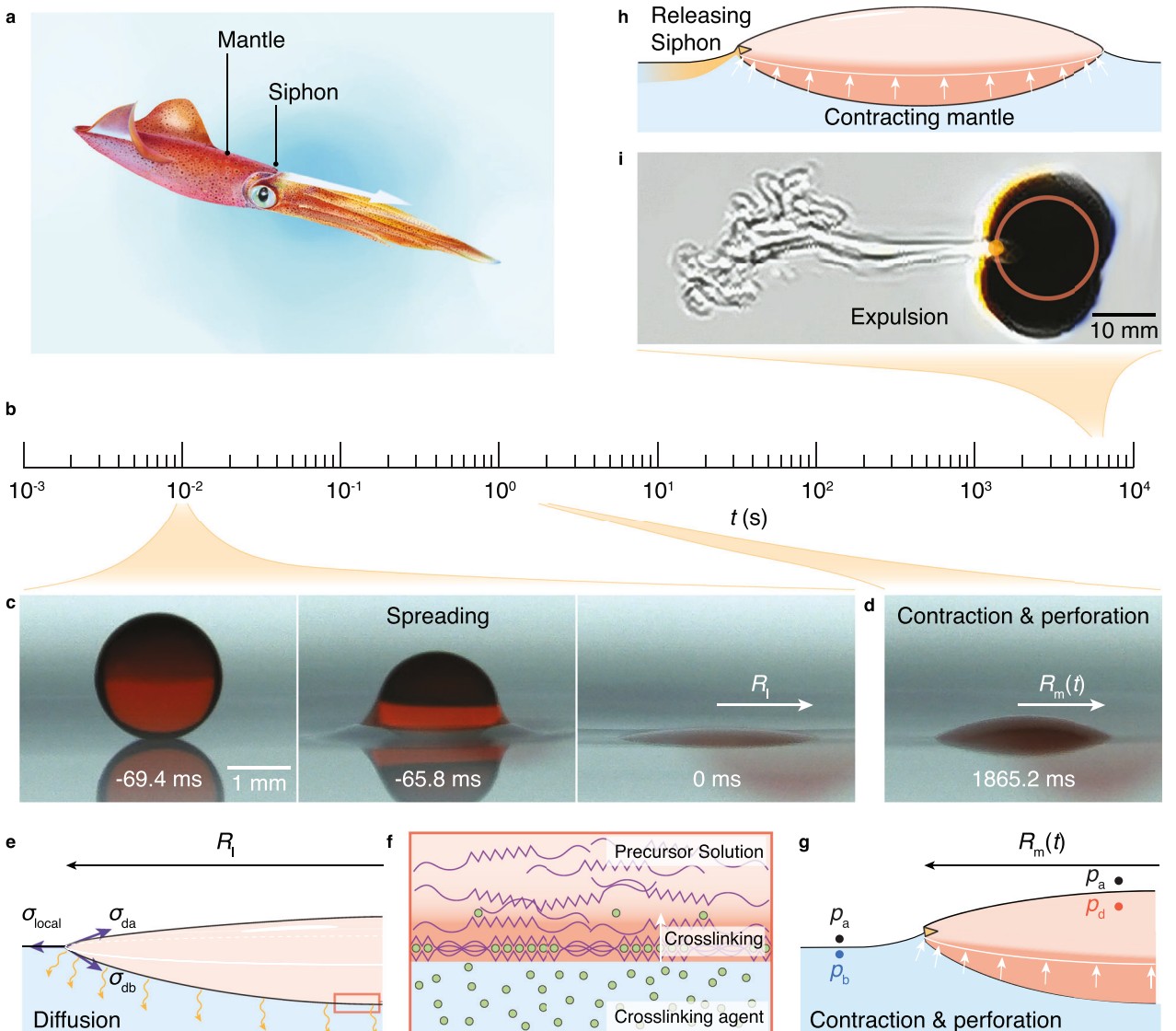

**Fig. 1 | Contracting mantle and siphon. a** Squid contracts the muscular mantle cavity to expel water through a narrow siphon for agile and sustained locomotion. **b** Timescales for generation and propulsion of the gelling-droplet motor. **c** When touching the bath solution, the droplet laden with surfactants spreads into a lens of finite radius $R_l$ in 69.4 ms. **d** Because of crosslinking, the lens radius reduces to $R_m$ at 1.86 s whereupon it initiates its locomotion. **e** Upon touching the calcium chloride solution bath, a droplet of alginate and surfactant mixture spreads into a liquid lens of finite radius $R_l$. Across the liquid interface, active molecules freely diffuse from the droplet into the bath. **f** Diffusion of calcium ions from the bath to the droplet triggers the chelation, forming the gel mantle, which significantly cuts the rapid surfactant diffusion. The crosslinking starts at the droplet-bath interface and advances toward the droplet bulk solution. **g** The gradual crosslinking lowers droplet radius $R_m(t)$, increasing the Laplace pressure of the liquid interior. Pressure buildup swiftly perforates the gel mantle, forming a siphon through which surfactant is released. **h** The gelling-droplet motor recapitulates the contracting-mantle-siphon propulsion mechanism of squid. **i** Schlieren visualization shows buffered jetting of the surfactant. Motor propulsion generates a wave around it, which distorts its image (see Supplementary Movie 5 for such image distortion when the motor initiates its motion). The red circle denotes the motor profile. Image in (**a**) is printed with permission from Zemin Chen.

Supplementary Fig 1 and Supplementary Movie 1). Note that the highest lifetime we measured is 194 min for a 35-μl droplet.

We find that such a simple gelling system autonomously recapitulates the squid's contracting-mantle-siphon configuration, producing a buffered surfactant jet for sustained propulsion. As shown in Fig. 1c, e, upon touching the solution bath, the surfactant-loading droplet spreads into a liquid lens, increasing capillary forces for its flotation[9]. Otherwise, in the absence of surfactant, the alginate-solution droplet sinks in the bath because of its slightly higher density (1.001 g ml⁻¹) and then crosslinks into a gel sphere (Supplementary Movie 3). The persistent gelation forms a contracting hydrogel mantle on the droplet, which cuts surfactant diffusion and squeezes the liquid interior, swiftly forming a siphon by perforating gel (Fig. 1d, f, g)[10]. Pumping by Laplace pressure, active molecules are unidirectionally released through the siphon whose

rate is well-buffered by a prominent viscous force (Fig. 1h, i). In this way, surfactants are released in an anisotropic and controlled manner, a state that maximizes the localized concentration inhomogeneity and prolongs adsorption saturation on the water surface, two effects critical to the efficiency and lifetime of Marangoni motors. Such a sustained artificial "squid" is self-generated and will be of interest and relevant to fields such as cargo delivery, active matter assembly, and microscale robotic powering.

## Results

### Short-timescale mantle-siphon generation
We start by using a commonly seen gelling pair, that is, sodium alginate and calcium chloride (Supplementary Table 2), which was first used by Ender et al. for micro-swimmers[11-14]. When a sodium alginate droplet is

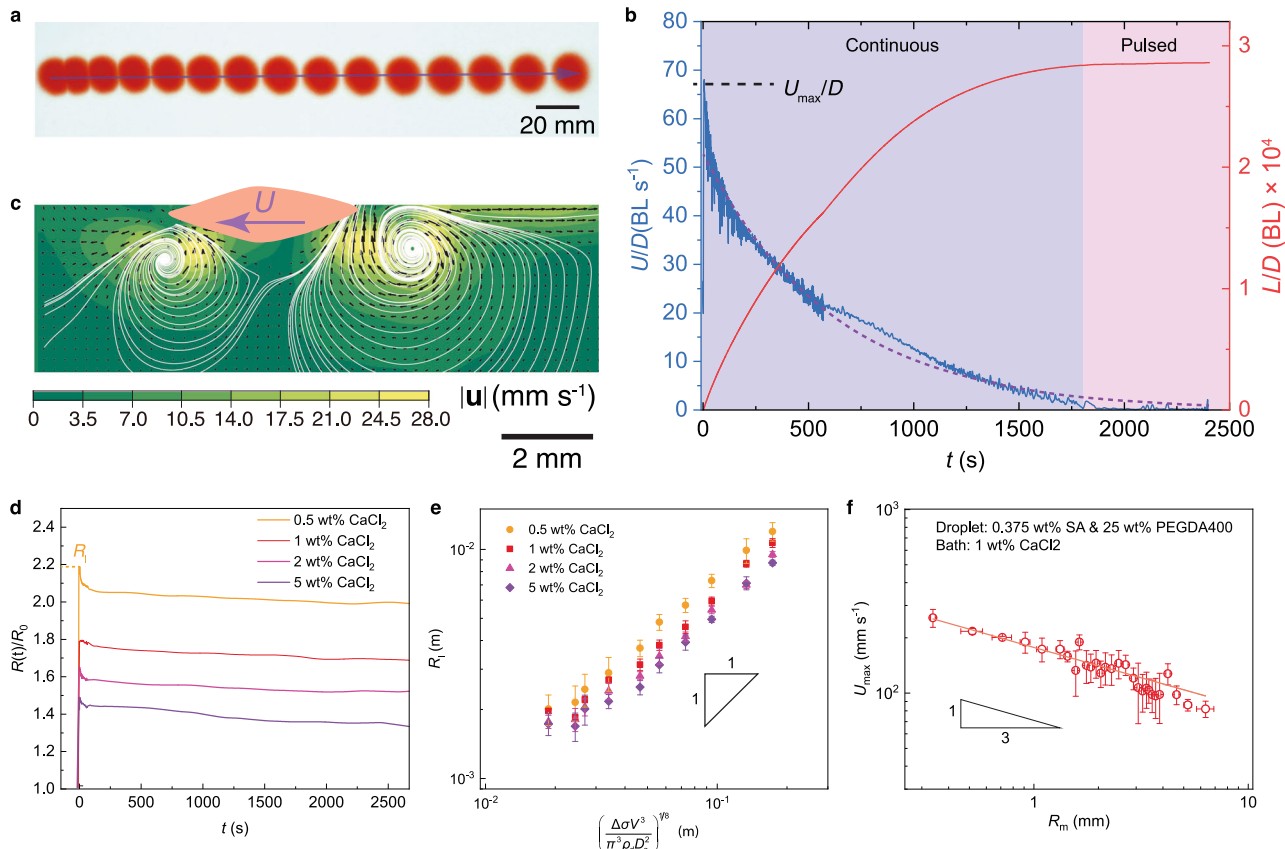

**Fig. 2 | Propulsion dynamics. a** Chronophotography (timestep of 100 ms) of a gelling droplet self-propels on the bath surface. The droplet is dyed red using direct red 23 for visualization. Because of many free hydroxyl and carboxyl groups, the sodium alginate linear chain strongly interacts with direct red dye molecules through hydrogen bonding. In this way, red dyes are trapped inside the motor[15]. **b** The distance and propelling velocity normalized by motor diameter $D$ as a function of time. Purple and magenta shadings, respectively, denote continuous and pulsed motion mode. Purple dashed line denotes the exponential fitting line, $U/D = 53e^{-0.0016t}$, for the propelling velocity after the initial 5 s. **c** Side-view PIV measurements are performed in the central plane of a propelling droplet. The unbalanced Marangoni stress advects asymmetric boundary-layer vortices. **d** Temporal evolution of the droplet radius $R(t)/R_0$. The moment when the droplet touches the bath is defined as $t = 0$. Color denotes the mass percentage of calcium chloride in the liquid bath. The droplet is an aqueous solution of 1 wt% sodium alginate and 25 wt% PEGDA400. **e** Logarithmic representation of the measured finite spreading radius $R_l$ and its scaling prediction. Error bars denote the standard deviation of 5 experiments. Color denotes the mass percentage of calcium chloride in the liquid bath. The droplet is an aqueous solution of 1 wt% sodium alginate and 25 wt% PEGDA400. Similar to ref. 16, an intercept appears here although no pre-factor is shown in Eq. 1. Such an intercept is potentially due to simplification in the assumption of spreading cessation and estimation of diffusion coefficient. **f** The maximum propulsion velocity $U_{max}$ as a function of the motor radius $R_m$. The red line is the fitting line. Error bars denote the standard deviation of 5 experiments. The droplet is an aqueous solution of 0.375 wt% sodium alginate and 25 wt% PEGDA400. The liquid bath is 1 wt% calcium chloride aqueous solution.

placed in a calcium chloride solution bath, it sinks and gels into a calcium alginate sphere. By contrast, when amphiphilic molecules are added to the droplet, it spontaneously spreads and floats on the bath surface in ~10 ms, followed by a rapid propulsion in ~1 s which continues for ~100 min (Figs. 1c and 2a, b). Note that the droplet is dyed red using direct red 23 for visualization. Red dyes are trapped inside the droplet as the sodium alginate chain strongly interacts with the dye molecules through hydrogen bonding[15]. For a 35 µl droplet containing 1-wt% sodium alginate and 25-wt% polyethylene glycol diacrylate (PEGDA) of an average molecular weight of 400, the droplet propels for 194 min at an average velocity of ~10 mm s⁻¹, an unexpectedly long lifetime affected by the short-timescale dynamics which we now investigate (Supplementary Movie 2).

As shown in Fig. 1c, d and Supplementary Movie 3, the lens radius captured through high-speed photography varies in a non-monotonic manner. When the droplet of a surface tension $\sigma_{da} = 42$ mN m⁻¹ touches the bath of a surface tension $\sigma_{ba} = 70$ mN m⁻¹, where $\sigma$ is surface tension, and subscripts d, b, a, respectively, denote droplet, bath, and air unless otherwise specified, it instantaneously spreads into a lens of a finite maximum radius $R_l$ in ~10 ms. The surface-active molecules

isotropically diffuse across the liquid interface without any barrier (Fig. 1e), a symmetric transport evidenced by the circular region of zero velocity mapped by the particle image velocimetry (PIV) on the surface (Supplementary Fig 2a). Note that the region of zero velocity forms as the Marangoni stress sweeps the tracer particles away, signifying the release of surfactant.

The Marangoni advection near the lens boundary lowers the local bath-air surface tension, causing the spreading coefficient $S \approx \sigma_{local} - (\sigma_{da} - \sigma_{db})$ to be roughly zero whereupon the spreading ends (Supplementary Note 1). Such process is similar to that of alcohol droplet spreading on water surface whose scaling analysis can be used to predict the $R_l$ as follows[16]:

$$R_l \sim \left( \frac{\Delta\sigma V^3}{\pi^3 \rho_d D_s^2} \right)^{1/8} \tag{1}$$

where $V$ is the droplet volume, $\rho$ is the mass density, $D_s$ is the surfactant diffusion coefficient. As shown in Fig. 2e, results largely fit well with the prediction but deviate when droplet volume is less than 6.94 µl.

Upon the charge-compensation crosslinking between carboxyl groups of alginate molecule chains and calcium ions, previously free and dissociated alginate chains aggregate and interconnect, forming a gel of reduced volume[17,18]. In this way, the lens contracts during the crosslinking of calcium alginate (Fig. 2d). The shrinking radius of the hydrogel mantle increases the curvature $\nabla \cdot \mathbf{n}$ of the atop droplet, where $\mathbf{n}$ is the surface unit normal vector, building the Laplace pressure $\sigma_d \nabla \cdot \mathbf{n}$ up. Note that throughout the lifetime, the motor's top remains in a liquid state as the upward diffusion of calcium ions is too slow to crosslink the entire droplet. In this way, the pressure inside the droplet can be well described using the interface curvature (if the top was crosslinked, stress in gel film would impact the droplet pressure). In Fig. 1g, as the length scale is much smaller than the capillary length, the hydrostatic pressure can be neglected. Beneath the bath-air interface, the pressure in liquid bath $p_b$ equals to the ambient pressure $p_a$. By contrast, pressure inside droplet $p_d$ is $p_a + \sigma_d \nabla \cdot \mathbf{n}$. Once the siphon forms, surfactant is pumped out because of higher $p_d$ (Supplementary Note 1). Because there is no strong pressure gradient inside the droplet, the curvature is largely uniform along the liquid-air interface.

Along the three-phase contact line, Laplace pressure and surface tension force (surface tension of bath-air interface perpendicularly acting along droplet perimeter) radially stretch the newly-crosslinked gel film, making a stress peak at mantle's perimeter (Supplementary Fig 3). When local circumferential tensile stress exceeds the gel strength, crack starts to form (Supplementary Fig 4). The pressure swiftly perforates the hydrogel mantle in ~1 s, forming a siphon for surfactant release whereupon the motor of a radius $R_m$ initiates its rapid translation (Figs. 1c and 2a).

Unlike the isotropic diffusion, such advected discharge is anisotropic (Supplementary Movie 4). During propulsion, the region of zero velocity becomes asymmetric (Supplementary Fig 2b). As shown by the releasing jet in the top-view Schlieren image in Fig. 1i, the active molecules are ejected through the siphon of ~$0.1R_m$ in size at the rear of the advancing motor, whereas circumferential rapid diffusion across the interface is largely cut by the gelled mantle. Such anisotropic and directional surfactant discharge maximizes the localized concentration inhomogeneity, substantially prolonging the lifetime.

## Long-timescale propulsion

Then we investigate the longer-timescale propulsion (~1000 s). As shown in Fig. 2b, when the motor initiates its locomotion, it rapidly reaches the maximum velocity $U_{max}$ in ~5 s. For subsequent propulsion in ~1000 s, its velocity decreases in an exponential manner, suggesting surfactant release is the first-order type wherein the releasing rate is proportional to the remaining amount (see details in Supplementary Note 1). Approaching the lifetime, the continuous locomotion becomes pulsed. Such pulsed motion is caused by an intermittent jet containing surface-active agent (Supplementary Fig 5 and Supplementary Movie 5). Therefore, even approaching the end of lifetime, that is, the pulsed mode, the jet accompanies the motor motility. As seen in Supplementary Fig 6 and Supplementary Movie 6, after motor forms and propels in 30 s, we terminate its further crosslinking by transferring it into water. Unlike the continuous-crosslinking one, it has a larger radius, which suggests that the crosslinking and thus slow shrinking continue throughout self-propulsion. Moreover, the continuous-crosslinking motor has a lifetime fourfold higher than that of the terminated one. We propose that as crosslinking proceeds, a thicker mantle forms, which prevents surfactant diffusion across the gel film more effectively.

The side-view PIV measurements unfold asymmetric counter-rotating vortices near the boundaries of a propelling motor (Fig. 2c and Supplementary Movie 4). Such a circulating flow is induced by the viscous shear of the Marangoni stress. A stronger rear vortex indicates higher Marangoni stress and thus faster surfactant release. The vortex

convects fresh bulk solution to the surface and assists in breaking the surfactant adsorption saturation on the bath surface[16].

The surfactants we used are soluble, and some are even volatile. Nevertheless, for simplification, here we assume that the surfactant is insoluble and nonvolatile so that the evaporation and the evolution of bulk surfactant concentration can be neglected[19]. The surface concentration of surfactant $\Gamma(x, y, t)$ evolves according to the advection-diffusion equation as follows:

$$\frac{\partial \Gamma}{\partial t} + \nabla_s \cdot (\Gamma \mathbf{u}_s) = S_{boundary} + D_s \nabla_s^2 \Gamma \qquad (2)$$

where $\nabla_s$ is the surface gradient operator, $\mathbf{u}_s$ is the surface velocity, $\nabla_s^2$ is the Laplacian, the implicit $S_{boundary}$ is the boundary source, which is set as a boundary condition in the numerical study. Given $D_s \sim 10^{-10}$ m$^2$ s$^{-1}$, we have a Péclet number Pe $\equiv R_m U/D_s \sim 10^6$, suggesting the advection dominates the transport (Supplementary Note 2).

We then simply assume that in a finite time $\Delta t$, surfactants are homogeneously deposited on the liquid surface swept by the droplet translating at a velocity $U$, providing $\Gamma = J/(0.1R_m U)$, where $J$ is the surfactant releasing rate (Supplementary Fig 7b). To verify the simplification, we numerically study the motor motion and evolution of surfactant concentration distribution through finite-element analysis using COMSOL-Multiphysics (see details in Supplementary Note 3 and Supplementary Movie 7). As shown in Supplementary Fig 7, when the motor propels, around its perimeter, the released surfactant is concentrated in the vicinity of releasing siphon. The simplification (pulse concentration distribution) is roughly consistent with such a distribution feature. Such simplification gives a Marangoni force of $F_m \sim \kappa \Gamma \cdot 0.1R_m = \kappa J/U$ where $\kappa \equiv -d\sigma/d\Gamma$.

During the propulsion, the Reynolds number is Re $\equiv \rho_b U R_m / \mu_b \sim 10^2 - 10^3$, implying a resistant force associated with the pressure distribution on the motor surface as $F_d = \rho_b U^2 C_d \pi R_m h_m$, where $C_d$ is the drag coefficient and $h_m$ is the height of the motor. A balance between the driving Marangoni force and the drag force provides the $U$ as follows:

$$U \sim \sqrt[3]{\frac{\kappa J}{\rho_b C_d \pi R_m h_m}} \qquad (3)$$

By substituting the first-order release assumption $J \sim c_m Q_0 e^{-kt}$ into Eq. 3, we have the temporal evolution of propulsion velocity as $U(t) \sim \left(\frac{\kappa c_m Q_0}{\rho_b C_d \pi R_m h_m}\right)^{1/3} e^{-kt/3}$ where $U_{max} \sim \left(\frac{\kappa c_m Q_0}{\rho_b C_d \pi R_m h_m}\right)^{1/3}$. As shown in Fig. 2f, experiments are roughly consistent with the derivation. Note that the scaling in Fig. 2f is semi-quantitative as $R_m$ and $U_{max}$ span for only one order of magnitude, which is the largest range that we can experimentally obtain.

## Generality and performance

To coax the simple gelling system into artificial squid propulsion, two constituents, that is, the hydrogel matrix and surfactant, should be well chosen. The hydrogel can work as a mantle if it gels fast and has sufficient mechanical strength to support a droplet. To show that, we utilize two other possible gelling pairs, sodium carboxymethyl cellulose (CMC-Na)/calcium chloride and chitosan/sodium hydroxide, as motor mantle. Similar to sodium alginate/calcium chloride, CMC-Na/calcium chloride crosslinks in an electrostatic manner. But chitosan/sodium hydroxide gels through hydroxide-induced deprotonation of amines. Using PEGDA400 as active molecules, all the chosen gelling pairs host sustained propulsion (Fig. 3a, see details in Supplementary Figs. 8–10). The CMC calcium motor lasts the longest (3214 s, 20 m), followed by calcium alginate (2971 s, 71 m) and chitosan (2182 s, 24 m) ones.

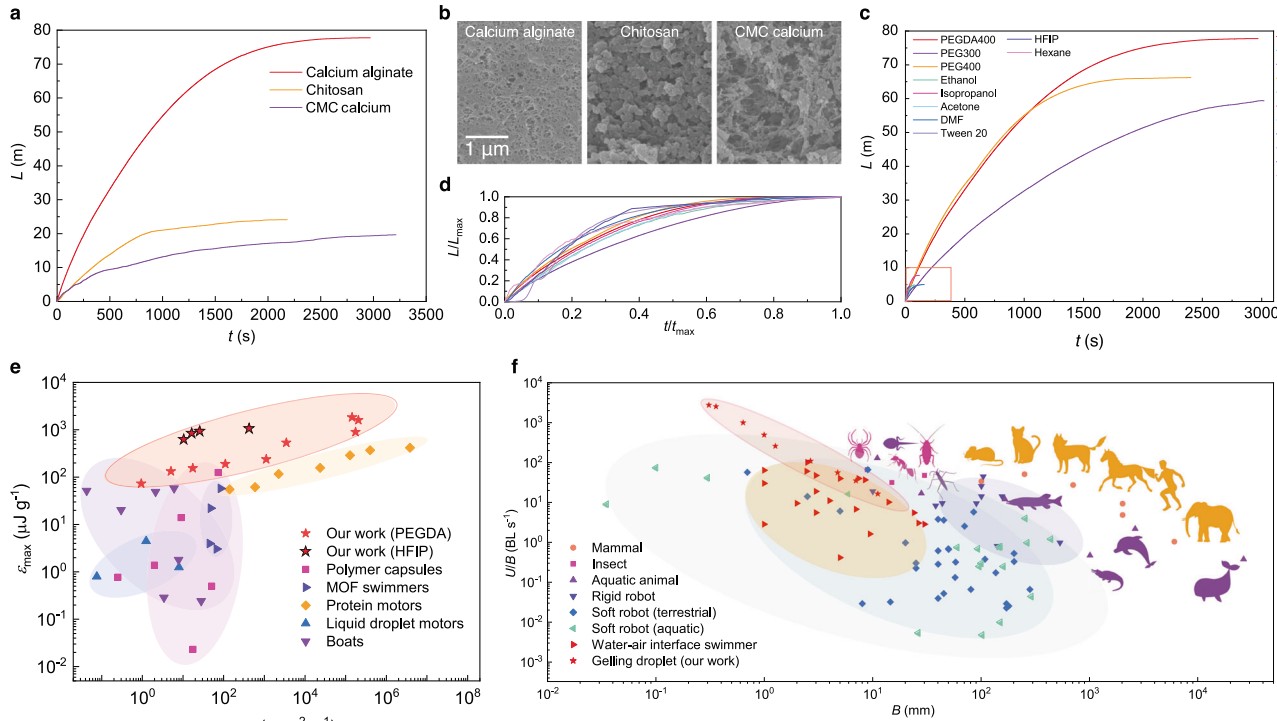

**Fig. 3 | Generality and performance. a** Sustained propulsion of gelling-droplet motor hosting by different hydrogel mantles. **b** Scanning electron microscopy images showing microscale structure of hydrogel mantles. **c** Different surfactants feature distinctive lifetimes. The non-small-molecule surfactant (low-molecular-weight polymer) sustains propulsion for ~1000 s, a value two orders of magnitude longer than that of the small-molecule surfactant (~10 s) which is highlighted in the red box and magnified in Supplementary Fig 8f. **d** The temporal positions of different surfactants roughly collapse onto a single nondimensional curve, implying the same propulsion mechanism. **e** Motor benchmark with respect to maximum performance output $\alpha_{max}$ and efficiency $\varepsilon_{max}$. Full references are given in Supplementary Table 5. **f** Logarithmic representation of the maximum relative velocities $U/B$ and size $B$ for insects, terrestrial and aquatic animals, artificial robots, and water-air interface swimmers. The light violet, blue violet, green, orange, and red shadings, respectively, denote rigid robot, terrestrial soft robot, aquatic soft robot, water-air interface swimmer, and our gelling-droplet motor. Full references are given in Supplementary Table 6.

As shown in Fig. 3b, hydrogel matrices are inherently porous, limiting the surfactant to be non-small molecules; otherwise, small molecules can quickly diffuse through the nanoscopic pores, leading to fast fuel depletion and reduced concentration non-uniformity. In Fig. 3c, we compare the motor lifetime loaded with different surfactants. The lifetime of small molecule surfactants such as ethanol, isopropanol (Supplementary Tables 3, 4) is limited to be ~100 s. By contrast, the lifetime of non-small-molecule polymer surfactant is ~1000 s. Despite such differences, their normalized temporal positions roughly collapse onto a single non-dimensional curve (Fig. 3d), suggesting that the contrastive kinematics have similar dynamics.

Along with high $U_{max}$ and long lifetime, the translational performance materialized through such a contracting mantle-siphon is high in view of other aspects. We benchmark the motor performance using two typical metrics for chemical motors, that is, output $\alpha_{max} \equiv U_{max}/V$ (maximum speed per unit volume) and efficiency $\varepsilon_{max} \equiv mU_{max}^2/2m_f$ (maximum kinetic energy per unit mass of surfactant), where $m$ is the initial mass of motor and $m_f$ is the total mass of surfactant loaded in the motor. These two metrics are used by Grzybowski et al. and Sitti et al. (Fig. 3e, Supplementary Table 5)[20,21]. The output of our motors can be as high as $2.06 \times 10^5$ mm$^{-2}$ s$^{-1}$ which is on a par with the highest reported value thus far. The efficiency leapfrogs to a high value, 1830 µJ g$^{-1}$.

In Fig. 3f, we parameterize the locomotion of natural creatures and artificial motors with respect to body length $B$ and relative velocity $U/B$ (Supplementary Table 6). The natural and artificial locomotion, regardless of terrestrial or aquatic type, shows a clear inverse relationship between relative velocity and body length, potentially owing

to mechanical constraints. Among these, our motors have a steeper slope, featuring an exceptionally high relative velocity of 2884.7 BL s$^{-1}$ for the motor of 0.15 mm in radius.

## Interfacial powering source

To explore potential applications and inspired by the work of Yu et al., we utilize our gelling-droplet motor to power their wireless and battery-free ion sensing system for real-time water quality monitoring[22]. Their system consists of sensing, data transmission, and propelling modules[22] (Fig. 4a and Supplementary Fig. 11). For battery-free and wireless sensing and data transmission, potentiometric sensors and near-field communication (NFC) are used in the system[23] (Fig. 4b). Our chemical motor is then used to actuate the motion of the sensing system without any connected cables.

We integrate 4-electrodes (3 ion-selective electrodes and 1 reference electrode) ionic sensors targeting 3 ions, Na$^+$, K$^+$, and NH$_4^+$, into the mobile electronic monitoring system[24] (Fig. 4c). The 3 ion sensors show a log-linear voltage-concentration relationship predicted by the Nernst equation and have strong selectivity and anti-interference capability (Fig. 4d–f and Supplementary Fig. 12). A gelling-droplet motor is then attached to the system and brings it into untethered motion in a confined circular water channel (Fig. 4g, h). By dripping a gelling-droplet motor of only 12.7 µl in volume, the sensor system is continuously propelled for nearly 30 min which shows that the gelling-droplet motor can work as simplified power module for interfacial machines (Fig. 4i and Supplementary Movie 8).

To make the motor perform macroscopic tasks apart from simple translation, we construct elementary interfacial machines

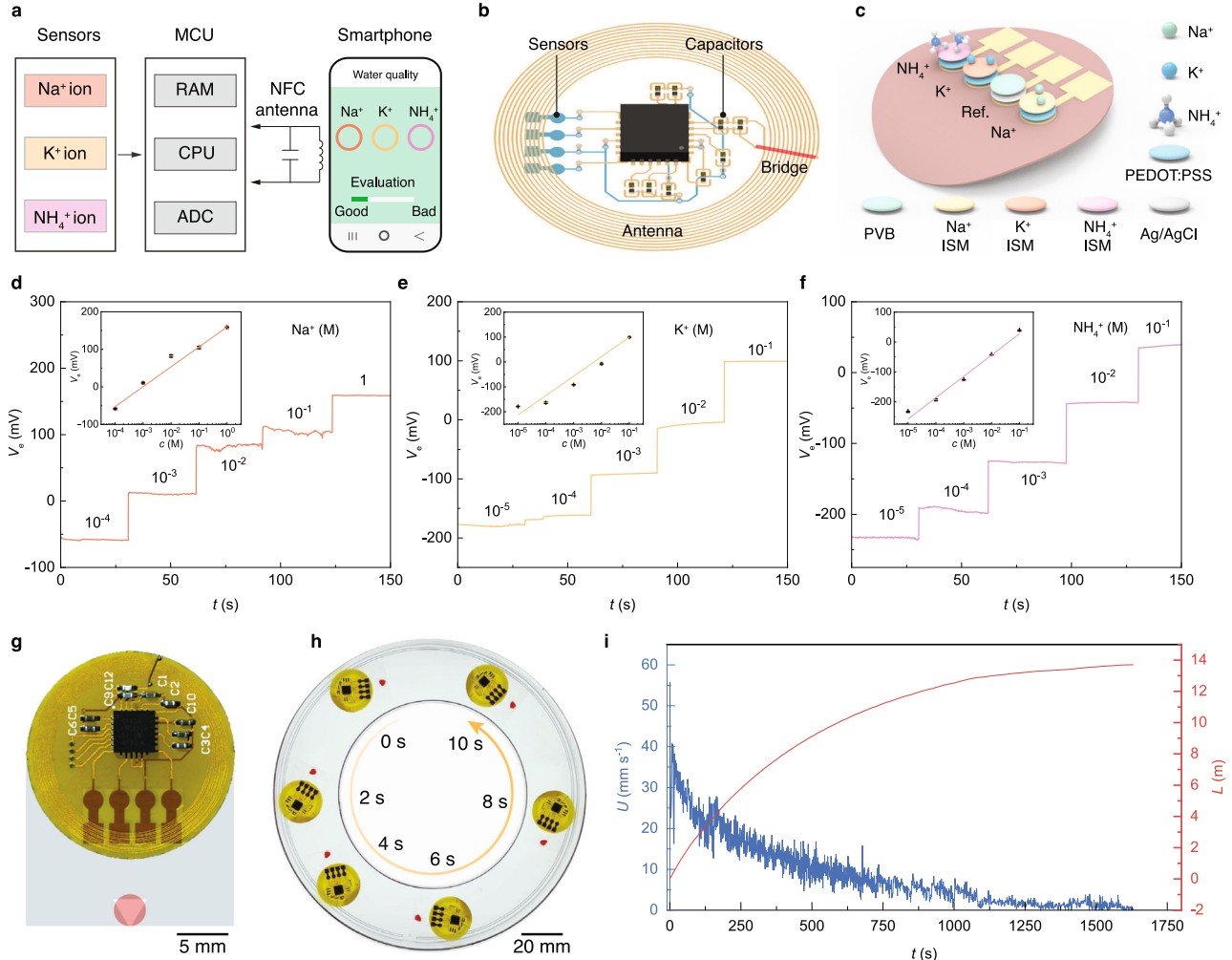

**Fig. 4 | Interfacial powering of a sensing device. a** Circuit logic diagram of the wireless sensing system. **b** Schematics showing the sensing and wireless transmission modules of the system. **c** Schematics showing a 4-electrode potentiometric sensor targeting for Na⁺, K⁺, and NH₄⁺ ions. **d** Potential-concentration response for Na⁺ ion sensor. Inset shows a log-linear relationship between potential signal and Na⁺ ion concentration. **e** Same measurement as in (**d**) for K⁺ ion. **f** Same measurement for NH₄⁺ ion. **g** Integration of wireless sensing system and gelling-droplet powering source. **h** Untethered locomotion of the sensing system in confined channel powered by a gelling droplet. **i** Sustained powering for the sensing system through a gelling droplet of only 12.7 µl in volume.

such as gear, cam, and crank, and construct them into transmission mechanisms (Fig. 5a-e)[25,26]. The interfacial machines are made by polyvinyl chloride sheet, allowing them to float on bath surface. As shown in Fig. 5a and Supplementary Movie 9, by depositing the motor on a gear tooth in an off-centre manner, the gear is driven into rotation whose direction is determined by the offset from the centreline in a radially outward direction. For example, the motor placed on the left side of a gear tooth will drive the gear into clockwise rotation. The rotation of the active gear can be transmitted to a passive gear (Fig. 5b).

Using such a mechanism (Supplementary Figs. 13–15 and Supplementary Movies 10, 11), we then construct the translational and swinging cam mechanisms whereby the cam rotation driven by the motor can be transformed into reciprocating rod translation or rocker swinging (Fig. 5c,d and Supplementary Movie 12). Note that in the state recovery stroke, the pushed rod or rocker restores their original position through lateral capillarity adhesion, producing intricate interaction between mechanical contact repulsion and short-range noncontact attraction. To enrich the system, a three-component mechanism is constructed in Fig. 5e and Supplementary Movie 13 wherein the crank rotation driven by motors is transformed into the rocker swinging through the intermediate linkage. The

reciprocating translation can be materialized by driving a slider inside a rail (Fig. 5f and Supplementary Movie 14).

## Discussion
Echoing the contracting mantle-siphon of a squid, we present a unidirectional and buffered fuel expulsion upon in situ crosslinking of a hydrogel droplet laden with surface-active molecules. The jet emanates out of a perforated siphon on the gelling mantle, an asymmetric geometry self-generated through pressure buildup. The mantle cuts isotropic diffusion, and its contraction generates a siphon through which surfactant is unidirectionally pumped. In this way, the localized inhomogeneity and adsorption unsaturation of surfactants are simultaneously upgraded, giving an unexpectedly long lifetime. Our approach, the crosslinking-mediated surfactant release achieved through a combination of chemicals, works for different gelation and surfactant chemical pairs and performs exceptionally well in terms of metrics such as output and efficiency. By integrating the Marangoni motor into an interfacial machine or engineering system, the chemical energy of the motor can be harnessed to do well-designed macroscopic work.

In this work, two classic but previously independent flow behaviors, interfacial spreading and microchannel flow, are linked

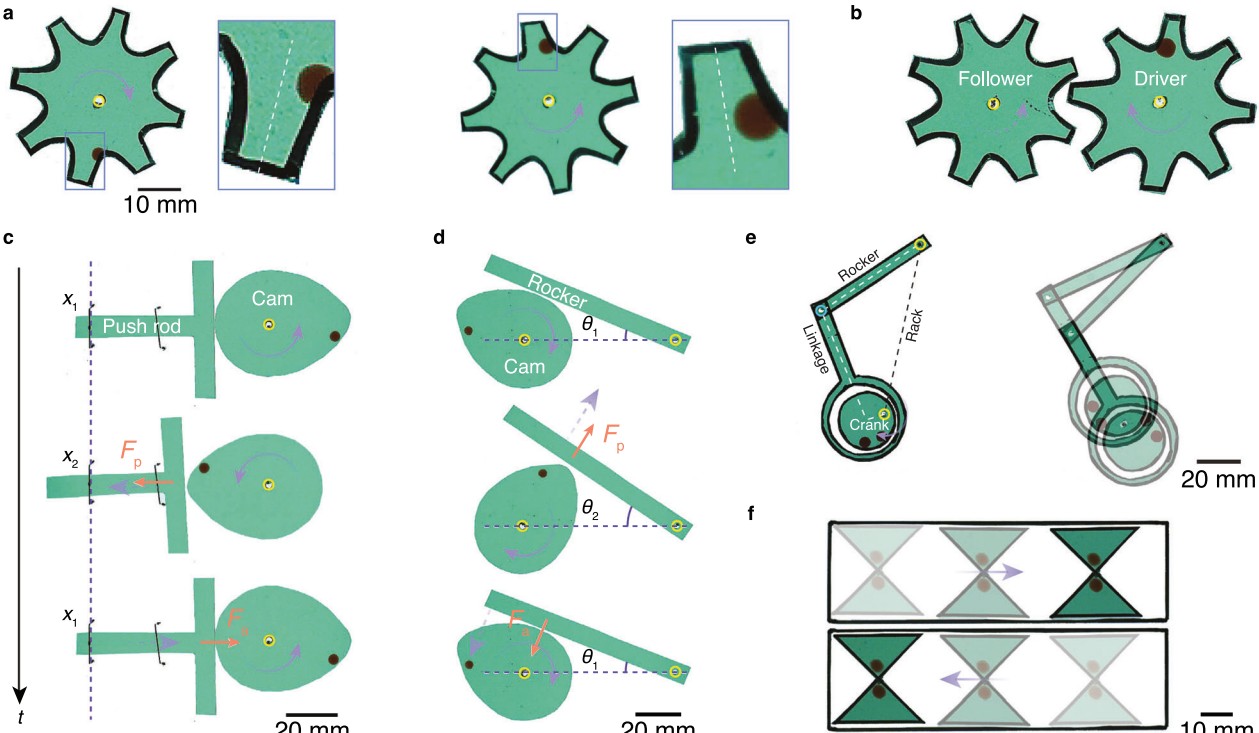

**Fig. 5 | Interfacial machine powering. a** Motors drive floating gears into clockwise (left) and counterclockwise (right) rotation. The off-centre deposition of the motor on the gear tooth determines the macroscopic rotation direction. **b** A gear system consisting of a driver and follower is driven by the motor. **c** Translational cam mechanism. The cam rotation driven by a motor is translated into linear reciprocating translation of the push rod. **d** Swinging cam mechanism. The cam rotation driven by a motor is translated into the reciprocating swinging of the rocker. Note that the restoring force acting on the push rod (**c**) and rocker (**d**) is the capillarity adhesion. **e** Crank-rocker mechanism. The rotation of the crank driven by a motor is translated into the swing of the rocker mediated by a linkage. **f** Two motors drive the reciprocating translation of a slider in a rail. Solid and dashed purple arrows, respectively, denote the motion of the driver and follower. Red arrows denote forces. The yellow circle denotes a fixed axle.

by a common-seen chemistry, chelation, generating an anisotropic fuel expulsion for sustained interfacial propulsion using a simple gelling system. Indeed, there are some issues that remain to be explored for this incipient approach. For example, control of rupture location on the contact line. The method holds potential to propel light-weight and floatable devices such as sensor, drug capsule, and even mini-generator[27,28] and will be attractive in environmental sensing[29,30], non-invasive treatment[31–33], microscale fabrication[34]. Such crosslinking-induced perforation may assist the study of accurate and effective delivery of medical drugs which are frequently encapsulated in hydrogel shells whose release is largely isotropic and diffusion-limited. The flow behavior may potentially be relevant to the burst and leakage of extracellular vesicles, which are critical to the intercellular communication, and thus the physiological and pathological cell function[35].

## Methods

The motor precursor solution is made by dissolving sodium alginate in 90-°C water with 1000-rpm magnetic stirring for 2 h, followed by surfactant addition at ambient temperature. The bath solution is made by dissolving calcium chloride in water. For motion tracking, the precursor solution is dyed red using direct red 23 (typically -0.5 to 1 wt%). To initiate the self-propulsion, precursor solution droplets are generated and released from a steel nozzle fixed atop the liquid bath surface at ~4 mm. Droplets' volume is controlled through the diameter of the nozzle.

Upon touching the bath solution, the instantaneous spreading and ensuing self-propulsion are recorded using high-speed photography at up to 5000 frames per second (Photron UX50). A single-mirror Schlieren optics setup is used to visualize the surfactant release.

For side-view PIV analysis, polystyrene microspheres of 50 μm in diameter are doped into the bath solution as tracer particles. For top-view PIV analysis, hydrophobic baby powder is seeded on the water surface as tracer particles. The velocity field is analysed using an open-source MATLAB code, PIVlab.

For interfacial machine powering, elementary machines of different geometries are fabricated by cutting a polyvinyl chloride sheet of 3 mm in thickness. Then the precursor solution is dripped at the edge of the machines for actuation.

The potentiometric ion-selective sensor is fabricated by patterning solid-state reference and working electrodes whose potential difference is measured to calculate ion concentration. The reference electrode is prepared by screen-printing Ag/AgCl ink onto a gold electrode. To minimize potential drift, a 2.5-μl methanol solution containing 79.1 mg ml$^{-1}$ polyvinyl alcohol butyral (PVB), 50 mg ml$^{-1}$ NaCl, 2 mg ml$^{-1}$ triblock copolymer poly(ethylene oxide)-poly(propylene oxide)-poly(ethylene oxide) (PEO-PPO-PEO, F127), and 0.2 mg ml$^{-1}$ carbon nanotubes is deposited onto the Ag/AgCl electrode. For the working electrodes, poly(3,4-ethylenedioxythiophene) polystyrene sulfonate (PEDOT:PSS) is electrochemically deposited onto three gold electrodes using an electrochemical workstation (CHI 660E) at a constant current of 141.36 μA for 1200-1800 s using a deposition solution containing 0.01 M 3,4-ethylenedioxythiophene (EDOT) and 0.1 M polystyrene sulfonate (NaPSS). Ion-selective membranes for Na$^+$, K$^+$, and NH4$^+$ are then deposited. The ionophore cocktail is a tetrahydrofuran solution containing 1.51 mg ml$^{-1}$ ionophore, 0.83 mg ml$^{-1}$ sodium tetrakis [3,5-bis(trifluoromethyl)phenyl] borate (NaTFPB), 50 mg ml$^{-1}$ polyvinyl chloride (PVC, K-value 72-1), and 99.17 mg ml$^{-1}$ bis(2-ethylhexyl) sebacate (DOS). The ionophores used for the Na$^+$, K$^+$,

and $NH_4^+$ ion sensors, respectively, are Na ionophore X, valinomycin, and $NH_4^+$ ionophore (nonactin).

## Data availability

The data supporting the findings of this study are available within the main text and the Supplementary Information. Any additional requests for information can be directed to and will be fulfilled by the corresponding author. Source data are provided with this paper.

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

## Acknowledgements

We thank Prof. Weiwei Deng, Prof. Huanshu Tan for equipment support. Prof. Bo Zhou, Prof. Canhui Yang for discussion. X.T. acknowledges funding from the National Natural Science Foundation of China (12588301), Shenzhen Medical Research Fund (A2303048), National Natural Science Foundation of China (12302354, 12572309), Shenzhen Science and Technology Program (JCYJ20220530114417040), and Southern University of Science and Technology (Y01646103).

## Author contributions

C.Z. and X.T. conceived and designed the project. C.Z. performed the experiments to which R.S., H.L., C.L., H.T., K.Z., and J.G. also contributed. H.T. and J.G. performed the numerical study. X.T., C.Z., H.T., and J.G. analysed the data. X.T., C.Z., H.T., and J.G. wrote the manuscript.

## Competing interests

The authors declare no competing interests.
