## [Transparent Peer Review file · Nature Communications]

Sustained Interfacial Powering through Self-Generated Mantle and Siphon of a Gelling Droplet

Corresponding Author: Professor Xin Tang

Version 0:

Reviewer comments:

Reviewer #1

(Remarks to the Author)

The authors present Marangoni-driven soft motors with extreme lifetimes using calcium alginate, carboxymethyl cellulose and chitosan polymers. The original idea is not new as soft motors have been published for calcium alginate (references 11-13) and chitosan (not referenced, <https://doi.org/10.1063/5.0097035>) but the novelty is the addition of another polymer (PEGDA) with which extreme lifetime can be achieved and the use as an interfacial sensor. Furthermore, from the motors machines performing more complex tasks than simple translation and rotation are created.

The manuscript is nicely built up using the analogy with the squid as "contracting-mantle-siphon configuration". The formation of the soft motors floating on solution is adequately characterized. The generality and the performance of the motors are excellently compared in Fig. 3 e and f with various artificial motors and animals (mammals, insects...).

The movies are of excellent quality, up to the point with correct lengths. The figures are of several subfigures but all needed and explained in the text. The SI contains the derivations but some are not sufficiently clear and the quantities yielding the dimensional numbers used should be also given.

Suggestions in detail:

- The amount/percentage of the added dye (direct red 23) should be added to the Materials section.
- Fig. 2: What is used for normalization in U/D in Fig. 2b, please specify.
- 2e: x axis should be shortened so the measurements are taking the larger part of the figure.
- 2f: y axis should be between 10^2 and 10^3 .
- Typo in Fig. 1 i: Expulsion is the correct word (not Explusion)
- Typo in Eqn. (3) - same as Eqn. S6: c_d should be C_d as this symbol was used as the drag coefficient above

Comments/suggestions related to SI:

- μ is dynamic viscosity, it should be changed throughout the text
- Re number: the relevant data from which $Re=100$ should be added (ρ , μ , R, and U)
- Derivation of S2 and S3 needs further explanation
- Ca number: relevant data or reference to the used data (μ_b , γ , U) from which Ca is between 0.01 and 0.1 should be added
- Pe number: relevant data or reference to the used data from which $Pe = 10^6$ should be added
- Gamma: probably the surfactant surface concentration, should be defined in the SI, the derivation of $\gamma = J_r / (2 R_m U)$ not clear, more explanation should be added
- typo in Eqn S6: c_d should be C_d as this symbol was used as the drag coefficient above
- Supp. Fig 2: Supp. Movie 3 is mentioned which is about gelation. What is the relevance to the figure. May be a typo?
- Supp Fig. 4 and 5 a and d: U_{max}/B based on the unit of $BL s^{-1}$ is probably a relative velocity for the body size. Is it correct? Please specify it in the caption.
- Sup Table 1: Density values should be centered throughout.
- Overall a list of abbreviation is very helpful for the readers, the addition of the units would be beneficial as well. Moving the list from the end of SI to the beginning would be better for the reader.

Based on above, I recommend the manuscript for publication after revision.

Reviewer #2

(Remarks to the Author)

The present manuscript tries to describe the underlying mechanism behind a self-propelling mantle and siphon swimmer created using a hydrogel-droplet combination. The manuscript also demonstrates a few applications of the aforementioned swimmer.

1. Mechanism of self-propulsion: My major concern with the present manuscript is that the mechanism of self-propulsion is poorly explained.

(a) To begin with, where and why does the hydrogel mantle perforate? The authors' explanation is that the shrinking hydrogel mantle results in increasing curvature of the floating lens, and hence increasing Laplace pressure, which eventually ruptures the mantle. What is the physics behind the shrinking hydrogel mantle? How is this related to the performance of the motor? Where is the location of the highest curvature? Is it at the three-phase contact line? What necessitates the rupture at that precise location? Does the hydrogel continue to shrink as the swimmer self-propels. Moreover, I failed to understand the shape of the swimmer from Fig. 1(i).

(b) I do not agree with the authors' conjecture that the surface tension gradient near the boundary of the swimmer is a constant (top of p. 6). The surfactant volumetric release rate is strictly not a constant over time since the liquid inside the hydrogel shell gets depleted over time. Moreover, if I understand correctly, equation S3 (in the supplementary material) is the reason for a source term in Eq. 2 (Eq. S4 in the supplementary material). If that is indeed the case, then the expression in equation S3 should be multiplied with some reference volumetric concentration of surfactant. Moreover, the relationship between Γ and J_r looks dimensionally inconsistent to me. I would request the authors to check this.

(c) The assumption of homogeneous surfactant deposition on the liquid surface swept by the swimmer is too simplistic, and almost borders on incorrect. This assumption overlooks the important fact that over the length scale of the swimmer (R_m) there has to be a finite gradient in the surfactant concentration in order to have self-propulsion. Ideally, Eq. 2 (or Eq. S4) needs to be solved along with the Euler's equations to obtain the distribution of the surfactant surface concentration since it is coupled to the high Reynolds number hydrodynamics.

(d) Unfortunately, I am not convinced with the data presented in Figs. 2(e) and (f). In 2(e), there are only 3 to 4 data points for each weight percentage, that too without any errorbars. Similarly, in 2(f), I am not convinced that the fitting line (I am guessing which represents Eq. 3) truly represents the experimental trend.

2. The definition of motor efficiency, and its subsequent calculation, are unclear.

3. In the conclusion, the authors' mention- "Our design works for different gelation and surfactant chemical pairs and performs exceptionally well in terms of metrics such as output and efficiency." It is unclear to me what are the 'design' parameters that are controllable apart from the choice of the chemicals.

4. The quality of the figures, specifically some of the graphs, leaves much to be desired.

To summarize, the manuscript fails to clearly explain and characterize the self-propulsion mechanism of the mantle and siphon droplet swimmer. In my opinion, the underlying physical mechanism is not rigorously explained in a quantitative manner to merit publication in Nature Communications. The applications demonstrated in the manuscript can only be useful to the broader scientific community once the underlying physical mechanism of the swimmer is fully grasped. Moreover, is it not possible for a simple camphor boat to perform these functions? Unfortunately, I cannot recommend the present manuscript for publication.

Reviewer #4

(Remarks to the Author)

The authors present a novel form of active soft matter which uses a combination of effects to produce truly fast (10mm/s), and long lived (~100minute) activity. This is enabled by the anisotropic crosslinking of a droplet which contains a surface active agent, and a cross linking polymer. As the polymer crosslinks, the surface active agent can only be released into the liquid as a perforation is formed as a result of anisotropic crosslinking. This technique is novel, and the effect it has as a combinatorial piece of active matter is phenomenologically rich. In this manuscript, the authors explore the mechanism with PIV, characterize other material systems which perform the same effect, and characterize the energetic limits with comparisons to literature. The authors also implement this system to push a water sensor, and to operate various examples of machines. This work is significant to the field, and the implementation of an actuation-marangoni soft matter robot is novel. The work largely supports its claims, while I have some concerns about a few details about the manuscript, listed below. The work is robust, and impressive, and after addressing the following concerns, including clarifying the mechanism and language, is suitable for publication.

Major:

- A central point of the manuscript is the relationship between the effect of the siphon, and the effect on the movement of the droplet, but this interaction is not well characterized when compared to the rest of the manuscript (which is robust). What does the perforation look like? Does the capsule perforate, or does the droplet simply not crosslink at one zone? Does the

"jet" of surface-active agent proceed throughout the experiment?

- Continuing in this direction, references to the importance of "Laplace" pressure occur frequently in the manuscript. The curvature does increase, but at the lengthscale of the droplet, the authors should demonstrate that it is the Laplace pressure that is moving surfactant out of the droplet.

Minor:

- The references and comparisons to the literature is somewhat incomplete, especially considering water-air interface swimmers and their introduction of anisotropy, and with reference to pulsing microrobots e.g., symmetry breaking.
- The authors refer to the system as a "host-guest" system, could they elaborate on why this system qualifies as a host-guest system?
- The authors claim that "Marangoni motor is plagued by short lifetime of ~10 s, limiting its practical application for modular powering source". It would be helpful to understand the intended object the proposed method would be used to power, and specifically what the physical limitations of traditional Marangoni-powered devices are.
- In paragraph 3 the authors state: "As shown in Figure 1c,e, upon touching solution bath, the surfactant-loading droplet spreads into a liquid lens, increasing capillary forces for its flotation⁹". What are the components of the droplet, is it more dense than water, and should I expect it to sink if not placed above the solution?
- The order, and depiction of figure 1 should be reconsidered. The connection to the various timescales of the experiment is an interesting way to depict the various behaviors of the droplet, but is confusing. To this end, the idea that the droplet is contracting when in contact with the solution (f) is very far away from 1h, where the importance of mantle contraction becomes apparent. Furthermore, indication of the crosslinking of the droplet-water interface would be helpful.
- The authors should consider using a phrase such as self-generated, self-directed as opposed to "malice of forethought" at the end of page 3, and should take a general look at the manuscript to clarify language in and around experiment, such as the word "liberating", or "lowers" rather than "weakens" surface tension.
- Figure 2a, the authors dyed the droplet red (alginate containing PEGDA), why is the droplet trapped in the PEGDA and not released?
- Supplementary movie 1: the authors compare a camphor boat to their own method to demonstrate the "long-lived" nature, can the authors provide a comparison of the "energy input" into either motor? What is the relative mass of the camphor, to the propellant in the viscous example? Why does the viscous example have a stated speed of ~100s mm/s, when the authors mention ~10s of mm/s in the introduction.
- Figure 2d/e/f: what is the wt% noted in the figure?
- Can the authors clarify what surface tensions they refer to, as in to which interface is this surface tension applied?
- The authors refer to a period of "pulsed motion", but the effect of the pulsed motion is not visible in the figure, or in any of supplementary videos. Can the authors include a note on the pulsed motion, the scale or reliability of pulsing, and a video?
- The perforated syphon should be shown in some capacity, either by microscopy, or post-experiment to demonstrate, or measure the size of the syphon.
- Does the top of the droplet gel? Does this gelling of the top of the droplet (if it does not occur) affect the performance of the droplet?

Version 1:

Reviewer comments:

Reviewer #1

(Remarks to the Author)

The authors revised their manuscript where they considered my suggestions and comments. The typos were corrected and the physical quantities are added when it was necessary.

The list of symbols in the SI is very useful.

The newly added movies are of high quality.

I accept all of the changes and recommend the manuscript for publication in the present form.

Reviewer #2

(Remarks to the Author)

I have now carefully gone through the authors' responses to my previous comments. I still have some questions, as mentioned below, which I think are critical to the scientific understanding of the work. I cannot recommend the publication of the present manuscript until these questions are unambiguously clarified.

1. I found the authors' arguments regarding the perforation of the hydrogel mantle to be hand-waving at best. If I understand the system correctly then there are two types of stresses acting on an elemental area of the hydrogel mantle-- one, a compressive stress due to the local Laplace pressure (this is the stress which is dependent on the curvature), and two, a tensile stress due the inherent tendency of the interfacial tension to minimize the surface area. If the perforation of the hydrogel mantle is dependent on the curvature (i.e. the shell perforates only once a critical shape is reached), then the compressive stress due to the Laplace pressure plays a dominant role in the rupture of the mantle. This is only my conjecture; the authors must explain it better. I think that even if the exact location of the rupture is not deterministic in nature, the mechanism of the rupture should be clearly explained. Otherwise, it will be impossible to engineer the system.

2. Supplementary Figure 5 (Figure R1): What does the constant value of the radius R_m represent, especially for the

micromotors in calcium chloride solution? If gelation continues, then I would expect R_m to be a function of time. In fact, the authors also consider the same during the evaluation of the liquid release rate from the micromotor (Eq. S3). Then what does the constant value of R_m represent in Figure S5/R1 (b) for a given concentration of calcium chloride? If $R_m(t)$ variation is insignificant as the authors claim, then the variation in the average curvature may also be insignificant. If that is the case, then why should the liquid release rate vary over time? More importantly, why should the speed U decay over time? Is $R(t)$ variation really negligible even for $t \sim O(10^2-10^3)$ s? Note that Figure 2(d) shows the variation only for the first 3 seconds. How would this variation look for $t \sim O(10^3)$ s? I would request the authors to show the decay of U over time in water, as well as in calcium chloride solutions of different concentrations.

3. Figure 2b (Figure R2): At present, the form of the exponential fit to the decay of U holds no physical justification. I would request the authors to investigate whether the co-efficient and the characteristic decay time can be estimated (at least the order of magnitude) from the explanation for the propulsion dynamics presented in section 2 of the supplementary material. Currently, 53 and 0.0016^{-1} have no physical meaning. Moreover, the right-hand side Y-axis should be L/D .

4. I am still confused with Fig. 2e. Equation S2 does not have a prefactor; therefore, in the log-log plot there should be no intercept. Right now for the plot to have slope 1, there has to be an intercept. Is there a physical genesis for the intercept?

5. The major concern with Fig. 2f is that the spread in the data is too great to reach any meaningful conclusion. In fact, on apparent inspection, a line with a higher slope seems to be a better fit than the line with slope $1/3$.

6. The definition of efficiency is still unclear. I did not understand what the authors mean by 'per unit mass of surfactant'. Can the authors please explain how they calculate this quantity?

Reviewer #4

(Remarks to the Author)

The authors have successfully addressed all of my comments, and suggestions. I believe that with the added data, explanations, i.e., with laplace pressure, and relevant literature, this manuscript is now suitable for publishing in Nature communications. The conclusions are now well-supported, and the work is of high quality.

Version 2:

Reviewer comments:

Reviewer #2

(Remarks to the Author)

I have gone through the authors' replies to my previous round of comments. I am satisfied with their replies. I think the manuscript has achieved the quality and rigour necessary for publication in Nature Communications. Hence, I recommend the manuscript for publication.

Response to Reviewers

Reviewer: 1

Comment 1:

The authors present Marangoni-driven soft motors with extreme lifetimes using calcium alginate, carboxymethyl cellulose and chitosan polymers. The original idea is not new as soft motors have been published for calcium alginate (references 11-13) and chitosan (not referenced, <https://doi.org/10.1063/5.0097035>) but the novelty is the addition of another polymer (PEGDA) with which extreme lifetime can be achieved and the use as an interfacial sensor. Furthermore, from the motors machines performing more complex tasks than simple translation and rotation are created. The manuscript is nicely built up using the analogy with the squid as "contracting-mantle-siphon configuration". The formation of the soft motors floating on solution is adequately characterized. The generality and the performance of the motors are excellently compared in Fig. 3 e and f with various artificial motors and animals (mammals, insects...). The movies are of excellent quality, up to the point with correct lengths. The figures are of several subfigures but all needed and explained in the text. The SI contains the derivations but some are not sufficiently clear and the quantities yielding the dimensional numbers used should be also given.

Our response: We sincerely thank the reviewer for his/her valuable and constructive comments, which have greatly contributed to improving the quality of our manuscript. We have revised the manuscript accordingly and provided the following point-by-point responses to each comment.

Our modification to the manuscript: In the Manuscript, we cited the paper of self-propelled chitosan hydrogel as Reference 14 (Page 14).

14. Kumar, P., Horváth, D. & Tóth, Á. Sol-gel transition programmed self-propulsion of chitosan hydrogel. *Chaos: An Interdisciplinary Journal of Nonlinear Science* **32** (2022).

Comment 2:

Suggestions in detail: -The amount/percentage of the added dye (direct red 23) should be added to the Materials section.

Our response: We thank the reviewer for this valuable suggestion. The amount of the dye is typically ~0.5 to 1 wt% which is added in the Materials and Methods section.

Our modification to the manuscript: In the Manuscript, we added the percentage of added dye in the Materials and Methods section (Page 16, Line 5).

Comment 3:

-Fig. 2: What is used for normalization in U/D in Fig. 2b, please specify.

2e: x axis should be shortened so the measurements are taking the larger part of the figure.

2f: y axis should be between 10^2 and 10^3 .

Our response: We thank the reviewer for these valuable suggestions to enhance the readability of the manuscript. In Figure 2b, velocity U was normalized by the motor diameter D . In Figure 2e, x-axis was shortened so the measurements take a larger part of the figure. In Figure 2f, y-axis was set between 10^2 and 10^3 .

Our modification to the manuscript: In the Manuscript, we added sentences in the caption of Figure 2b to specify that the propelling velocity U is normalized by motor diameter D (Page 21, Lines 6,7); we replotted Figure 2e so that data takes a larger part of the figure (Page 21); we shortened the y-axis of Figure 2f so that its range is from 10^2 to 10^3 (Page 21).

Comment 4:

-Typo in Fig. 1 i: Expulsion is the correct word (not Explusion)

-Typo in Eqn. (3) - same as Eqn. S6: c_d should be C_d as this symbol was used as the drag coefficient above

Our response: We sincerely thank the reviewer for giving us the opportunity to correct the typos. In Figure 1i, original word “*Explusion*” is corrected to be “*Expulsion*”. The symbol denotes for drag coefficient is revised to be C_d .

Our modification to the manuscript: In the Manuscript, we corrected the word from “*Explusion*” to “*Expulsion*” in Figure 1i (Page 19); we corrected the symbol for drag coefficient from “ c_d ” to “ C_d ” in Equation 3 (Page 8). In the Supplementary Materials, we corrected the symbol for drag coefficient from “ c_d ” to “ C_d ” in Equation S9 which is the original Equation S6 (Page 5).

Comment 5:

Comments/suggestions related to SI:

- μ is dynamic viscosity, it should be changed throughout the text
- Re number: the relevant data from which $Re=100$ should be added (ρ , μ , R , and U)
- Derivation of S2 and S3 needs further explanation
- Ca number: relevant data or reference to the used data (μ_b , γ , U) from which Ca is between 0.01 and 0.1 should be added
- Pe number: relevant data or reference to the used data from which $Pe = 10^6$ should be added

Our response: We thank the reviewer for giving us the opportunity to clarify the key details. Throughout the text, we clarified that μ is dynamic viscosity. In the Supplementary Materials, $Re = 100$ is calculated using $\rho_d \approx 10^3 \text{ kg m}^{-3}$, $u \approx 0.1 \text{ m s}^{-1}$, $R_l \approx 10^{-3} \text{ m}$, $\mu_d \approx 10^{-3} \text{ Pa s}$; $Re = 10^2$ - 10^3 is calculated using $\rho_b \approx 10^3 \text{ kg m}^{-3}$, $U \approx 0.1$ to 1 m s^{-1} , $R_m \approx 10^{-3} \text{ m}$, $\mu_b \approx 10^{-3} \text{ Pa s}$; $Ca = 10^{-2}$ - 10^{-1} is calculated using $\mu_b \approx 10^{-3} \text{ Pa s}$, $U \approx 0.1$ to 1 m s^{-1} , $\sigma \approx 0.01 \text{ N m}^{-1}$; $Pe = 10^6$ is calculated using $D_s \sim 10^{-10} \text{ m}^2 \text{ s}^{-1}$, $U \approx 0.1 \text{ m s}^{-1}$, $R_m \approx 10^{-3} \text{ m}$. All the data used was added in the Supplementary Materials. We added further explanation of Equation S2 and S3 in Supplementary Materials.

Our modification to the manuscript: Throughout the text, we clarified that μ is dynamic viscosity. In the Supplementary Materials, we added relevant data to calculate Re (Page 1, Lines 16, 17 and Page 5, Lines 11, 12), Ca (Page 4, Line 5), and Pe (Page 4, Line 10); we added explanation (Page 2, Lines 1-6 and 10 for Equation S2 and Page 3, Lines 1-6 for Equation S3) to detail the derivation of Equation S2 and S3.

Comment 6:

- Γ : probably the surfactant surface concentration, should be defined in the SI, the derivation of $\Gamma = J_r / (2 R_m U)$ not clear, more explanation should be added

Our response: We thank the reviewer for these valuable suggestions to enhance the readability of the manuscript. We defined Γ as surfactant surface concentration and added explanation to detail the derivation of Equation S7 in the Supplementary Materials.

Our modification to the manuscript: In the Supplementary Materials, we defined Γ as

surfactant surface concentration (Page 4, Line 2); we added explanation to detail the derivation of Equation S7 (Page 4, Lines 12-16).

Comment 7:

- typo in Eqn S6: c_d should be C_d as this symbol was used as the drag coefficient above

Our response: We sincerely thank the reviewer for giving us the opportunity to correct the typos. Throughout the manuscript, the symbol for drag coefficient is revised to be C_d .

Our modification to the manuscript: In the Supplementary Materials, we corrected the symbol for drag coefficient from “ c_d ” to “ C_d ” in Equation S9 which is the original Equation S6 (Page 5).

Comment 8:

- Supp. Fig 2: Supp. Movie 3 is mentioned which is about gelation. What is the relevance to the figure. May be a typo?

- Supp Fig. 4 and 5 a and d: U_{\max}/B based on the unit of $BL s^{-1}$ is probably a relative velocity for the body size. Is it correct? Please specify it in the caption.

Our response: We sincerely thank the reviewer for giving us the opportunity to correct the typos as well as provide essential details. We revised original “*Supplementary Movie 3*” to “*Supplementary Movie 4*” in the caption of Supplementary Figure 2 so that correct video is referred to. We added description in the caption of Supplementary Figure 8, 9 (original Supplementary Figure 4, 5) to specify that it is the relative velocity calculated by normalizing propelling velocity U by motor diameter D .

Our modification to the manuscript: In the Supplementary Materials, we revised original “*Supplementary Movie 3*” to “*Supplementary Movie 4*” in the caption of Supplementary Figure 2 (Page 10, Line 7); We added sentences in the caption of Supplementary Figure 8, 9, which are original Supplementary Figure 4, 5 (Page 16, Lines 3, 8, 9 and Page 17, Lines 3, 8) to specify that it is the relative velocity calculated by normalizing propelling velocity U by motor diameter D .

Comment 9:

- Sup Table 1: Density values should be centered throughout.
- Overall a list of abbreviation is very helpful for the readers, the addition of the units would be beneficial as well. Moving the list from the end of SI to the beginning would be better for the reader.

Based on above, I recommend the manuscript for publication after revision.

Our response: We sincerely thank the reviewer for pointing these out. We centred the density values in Supplementary Table 2 (original Supplementary Table 1). We added units in the list of symbols and moved the list to the beginning of Supplementary Tables so that it becomes Supplementary Table 1.

Our modification to the manuscript: In the Supplementary Materials, we centred the density values throughout in Supplementary Table 2, which is original Supplementary Table 1 (Page 25); we added units in the list of symbols and moved the list to the beginning of supplementary tables (Page 23, 24).

We would like to thank the reviewer once again for expending so much time and effort in providing a review which has helped us to improve the clarity and specificity of our manuscript.

Reviewer: 2**Comment 1:**

The present manuscript tries to describe the underlying mechanism behind a self-propelling mantle and siphon swimmer created using a hydrogel-droplet combination. The manuscript also demonstrates a few applications of the aforementioned swimmer.

Our response: We sincerely thank the reviewer for his/her valuable and constructive comments, which have greatly contributed to improving the quality of our manuscript, particularly in the mechanism section. We have revised the manuscript accordingly and provided the following point-by-point responses to each comment.

Comment 2:

1. Mechanism of self-propulsion: My major concern with the present manuscript is that the mechanism of self-propulsion is poorly explained.

(a) To begin with, where and why does the hydrogel mantle perforate? The authors' explanation is that the shrinking hydrogel mantle results in increasing curvature of the floating lens, and hence increasing Laplace pressure, which eventually ruptures the mantle.

Our response: We thank the reviewer for these insightful comments. Expansion pressure is applied across the mantle surface because of pressure difference between droplet and liquid bath. Apart from that, extra surface tension force acts along three-phase contact line and radially stretch the gel mantle. In this way, the stress tends to localize at motor's perimeter. When circumferential tensile stress in the perimeter exceeds the gel strength, the gel fractures. Because of inhomogeneity in gel strength or localized stress concentration, the exact perforation position along the perimeter cannot be precisely controlled. However, because of high tensile stress, the perforation preferentially occurs at the contact line.

Our modification to the manuscript: In the Manuscript, we added sentences (Page 6, Lines 2-5) to explain that the hydrogel mantle is perforated at the three-phase contact line when local circumferential tensile stress exceeds the gel strength. We added sentences (Page 12, Lines 18, 19) to acknowledge that the exact fracture position on the three-phase contact line cannot be controlled yet.

Comment 3:

(A) What is the physics behind the shrinking hydrogel mantle? (B) How is this related to the performance of the motor?

Our response: We sincerely appreciate the reviewer's insightful comments. To provide clearer responses, we have divided them into two sections.

(A) What is the physics behind the shrinking hydrogel mantle?

Our response: We thank the reviewer for the insightful comment. As alginate molecular chain crosslinks with calcium ions, those previously free chains aggregate and connect with each other, forming a tight network of reduced volume. Such volume-shrinking effect associated with the alginate gel is reported in Reference 17 & 18.

Our modification to the manuscript: To explain the physics behind the shrinking hydrogel mantle, in the Manuscript, we added sentences (Page 5, Lines 8-11) and Reference 17,18; in the Supplementary Materials, we added sentences (Page 2, Line 15).

17. Gong, J., Schuurmans, C. C., Genderen, A. M. v., Cao, X., Li, W., Cheng, F., He, J. J., López, A., Huerta, V. & Manríquez, J. Complexation-induced resolution enhancement of 3D-printed hydrogel constructs. *Nature communications* **11**, 1267 (2020).

18. Cao, P., Tao, L., Gong, J., Wang, T., Wang, Q., Ju, J. & Zhang, Y. 4D printing of a sodium alginate hydrogel with step-wise shape deformation based on variation of crosslinking density. *ACS Applied Polymer Materials* **3**, 6167-6175 (2021).

(B) How is this related to the performance of the motor?

Our response: We sincerely thank the reviewer for the constructive comment. As shown in Figure R1 and Supplementary Movie 6, we added experiments to examine the impact of shrinking hydrogel mantle on the performance of the motor. Micromotors are firstly formed in CaCl₂ liquid bath for ~30 s and then transferred into water whereupon crosslinking is terminated. Their radius R_m , lifetime t_{max} , and maximum velocity U_{max} are compared with those of ones maintained in CaCl₂ (crosslinking persists for the entire lifetime). The transferred ones have slightly larger radius, suggesting that the crosslinking and thus very slow shrinking continues throughout self-propulsion. Because the two groups have the same conditions in the very first 30 s, so they have basically the same maximum velocity. By contrast, the continuous-

crosslinking motor has a lifetime fourfold higher than that of the terminated one. We propose that as crosslinking proceeds, thicker mantle forms which prevents surfactant diffusion across gel film more effectively. Therefore, continuous crosslinking has impact on motor's performance. However, the size shrinking associated with the continuous crosslinking is very small. Therefore, in such long timescale, the size shrinking has limited impact on the motor performance.

Figure R1 (Supplementary Figure 5 in the Supplementary Materials). Continuous crosslinking during self-propulsion. (a) Micromotors are formed in CaCl₂ liquid bath for ~30 s and then transferred into water whereupon crosslinking is largely terminated (see Supplementary Movie 6). Comparison of micromotor's radius R_m (b), lifetime t_{max} (c), and maximum velocity U_{max} (d) for ones maintained in CaCl₂ (crosslinking persists for the entire lifetime) and ones transferred into water after ~30 s (crosslinking only persists for the first 30 s). Error bars denote standard deviation of 3 experiments.

Our modification to the manuscript: To explain how the continuous crosslinking relates to the performance of the motor, in the Manuscript, we added sentences (Page 7, Lines 2-8); in the Supplementary Materials, we added Supplementary Figure 5 and Supplementary Movie 6.

Comment 4:

Where is the location of the highest curvature? Is it at the three-phase contact line?

Our response: We sincerely thank the reviewer for the insightful comment. The curvature of

liquid-air interface signifies the pressure inside the droplet. Because there is no strong pressure gradient inside the droplet, the curvature is uniform along the liquid-air interface at droplet's top.

Our modification to the manuscript: In the Manuscript, we added sentences (Page 5, Lines 20 and 21; Page 6, Line 1) to explain that as no strong pressure gradient is present in droplet, therefore, the curvature is largely uniform.

Comment 5:

What necessitates the rupture at that precise location?

Our response: We sincerely thank the reviewer for this constructive comment. Along three-phase contact line (note that thin gel film forms at the droplet-bath interface), Laplace pressure and surface tension force (surface tension of bath-air interface perpendicularly acting along droplet perimeter) radially stretch the newly-crosslinked gel film. When local circumferential tensile stress exceeds the gel strength, crack starts to form. In this way, the rupture preferentially occurs at the contact line. The exact rupture location may be associated with sites of stress concentration or gel defect. In our incipient approach, we cannot control the exact rupture point along the contact line.

Our modification to the manuscript: In the Manuscript, we added sentences (Page 6, Lines 2-5) to explain that the rupture occurs at the three-phase contact line; we added sentences (Page 12, Lines 18 and 19) to acknowledge that we cannot control the precise rupture location along the three-phase contact line.

Comment 6:

Does the hydrogel continue to shrink as the swimmer self-propels.

Our response: We appreciate the reviewer's insightful comment. As shown in Figure R1 and Supplementary Movie 6, we added experiments to examine the continuous shrink throughout self-propulsion. After motor forms and propels in 30 s, we terminate its further crosslinking by transferring it into water. Unlike the continuous-crosslinking one, it has larger radius which suggests that the crosslinking and thus slow shrinking continues throughout self-propulsion.

Figure R1 (Supplementary Figure 5 in the Supplementary Materials). Continuous crosslinking during self-propulsion. (a) Micromotors are formed in CaCl₂ liquid bath for ~30 s and then transferred into water whereupon crosslinking is largely terminated (see Supplementary Movie 6). Comparison of micromotor's radius R_m (b), lifetime t_{max} (c), and maximum velocity U_{max} (d) for ones maintained in CaCl₂ (crosslinking persists for the entire lifetime) and ones transferred into water after ~30 s (crosslinking only persists for the first 30 s). Error bars denote standard deviation of 3 experiments.

Our modification to the manuscript: To verify that the droplet continues to shrink as the droplet self-propels, in the Manuscript, we added sentences (Page 7, Lines 2-5); in the Supplementary Materials, we added Supplementary Figure 5 and Supplementary Movie 6.

Comment 7:

Moreover, I failed to understand the shape of the swimmer from Fig. 1(i).

Our response: Thanks for bringing this to our attention. As motor propels, it creates water wave around it. Such uneven water surface distorts its image in the Schlieren visualization. As shown in Supplementary Movie 5, when the motor is static, it appears circular. Once it moves, its profile in Schlieren image is distorted.

Our modification to the manuscript: In the Manuscript, we added sentences in caption of Figure 1i (Page 20, Lines 6-8) to explain that the water wave caused by motion distorts the droplet's profile in the Schlieren image. In the Supplementary Materials, we added

Supplementary Movie 5 to confirm such image distortion.

Comment 8:

(b) I do not agree with the authors' conjecture that the surface tension gradient near the boundary of the swimmer is a constant (top of p. 6).

Our response: We thank the reviewer for giving us the opportunity to clarify our misleading description. We want to express that the vortex convects fresh bulk solution to the surface and assists to break the surfactant adsorption saturation on bath surface. The original misleading description has been revised accordingly.

Our modification to the manuscript: In the Manuscript, we rewrote the original misleading sentence *“The vortex assists to sustain a constant surface tension gradient near the boundary by convecting fresh bulk solution to the surface, a flow breaking the surfactant adsorption saturation on bath surface.”* into *“The vortex convects fresh bulk solution to the surface and assists to break the surfactant adsorption saturation on bath surface”* (Page 7, Lines 12 and 13).

Comment 9:

The surfactant volumetric release rate is strictly not a constant over time since the liquid inside the hydrogel shell gets depleted over time. Moreover, if I understand correctly, equation S3 (in the Supplementary Materials) is the reason for a source term in Eq. 2 (Eq. S4 in the Supplementary Materials). If that is indeed the case, then the expression in equation S3 should be multiplied with some reference volumetric concentration of surfactant.

Our response: We sincerely thank the reviewer for this valuable comment. For source term,

we derived that the liquid volumetric releasing rate out of the droplet is $Q \sim 10^{-3} \frac{R_m(t)^3 \sigma_d \nabla \cdot \mathbf{n}}{\mu_d}$

(Equation S3). For long-timescale propulsion (~ 1000 s), motor size R_m slightly shrinks, σ_d and μ_d can be considered constant, so variation of Q is controlled by $\nabla \cdot \mathbf{n}$. However, the temporal evolution of $\nabla \cdot \mathbf{n}$ is exceptionally challenging to experimentally capture for the rapidly moving millimetre motor. In accordance with the reviewer's suggestion, as shown in Figure R2, for long-timescale propulsion (~ 1000 s), droplet's velocity decreases in an exponential

manner, suggesting surfactant release is the first-order type wherein the releasing rate is proportional to the remaining amount. On the basis of such velocity evolution feature, releasing rate can be phenomenologically considered as the first-order release. Thus, we can simplify the liquid volumetric releasing rate as $Q \sim Q_0 e^{-kt}$ (Equation S4 in the Supplementary Materials), where Q_0 is the initial volumetric releasing rate $Q_0 = 10^{-3} \frac{R_m(t)^3 \sigma_d \nabla \cdot \mathbf{n}_i}{\mu_d}$ with $\nabla \cdot \mathbf{n}_i$ being droplet-air curvature upon motion initiation, k is the first-order releasing rate constant. In this way, surfactant releasing rate in the unit of mol s^{-1} is $J \sim c_m Q_0 e^{-kt}$ (Equation S5 in the Supplementary Materials), where c_m is the surfactant concentration in the droplet.

Figure R2 (revised Figure 2b in the Manuscript). The distance and propelling velocity normalized by motor diameter D as a function of time. Purple and magenta shadings, respectively, denote continuous and pulsed motion mode. Purple dashed line denotes the exponential fitting line, $y = 53e^{-0.0016x}$, for the propelling velocity after initial 5 s.

Our modification to the manuscript: In the Manuscript, we added an exponential fitting line for motor's velocity evolution in Figure 2b (Page 21) and sentences (Page 6, Lines 18-20) to highlight the potential first-order releasing feature associated with the velocity exponential decrease. We added sentences in the Supplementary Materials (Page 3, Lines 7-15) to detail the derivation of such first-order surfactant release.

Comment 10:

Moreover, the relationship between Γ and J_r looks dimensionally inconsistent to

me. I would request the authors to check this.

Our response: We thank the reviewer for giving us an opportunity to check and correct the terms in Equation 2. In the corrected Equation 2, the implicit source term S_{boundary} , which is set as a boundary source in numerical study, has a unit of $\text{mol m}^{-2} \text{s}^{-1}$. In this way, it has a physical dimension of $[\text{mol L}^{-2} \text{T}^{-1}]$, which is the same as that of the other terms in Equation 2 such as the change rate of surface concentration $d\Gamma/dt$.

Our modification to the manuscript: In the Supplementary Materials, we added units in the Table 1. List of Symbols (Page 23 and 24) to show that the source term S_{boundary} has a unit of $\text{mol m}^{-2} \text{s}^{-1}$ which is consistent with that of change rate of surface concentration $d\Gamma/dt$ in the Equation 2 (Page 7).

Comment 11:

(c) The assumption of homogeneous surfactant deposition on the liquid surface swept by the swimmer is too simplistic, and almost borders on incorrect. This assumption overlooks the important fact that over the length scale of the swimmer (R_m) there has to be a finite gradient in the surfactant concentration in order to have self-propulsion. Ideally, Eq. 2 (or Eq. S4) needs to be solved along with the Euler's equations to obtain the distribution of the surfactant surface concentration since it is coupled to the high Reynolds number hydrodynamics.

Our response: We thank the reviewer for this insightful comment. We numerically solved Equation 2 coupled with fluid dynamics and motor motions. As shown in Figure R3, we numerically study the motor motion and evolution of surfactant concentration distribution through finite-element analysis using COMSOL-Multiphysics. When the motor propels, around its perimeter, the released surfactant is concentrated in the vicinity of releasing siphon. Far behind motor, the released surfactant is diffused and spread out. In terms of Marangoni propulsion, the surface concentration in the vicinity of contact line matters as Marangoni force is the line integral of surface tension acting on motor's perimeter. Near the contact line, the simplified assumption (Equation S7, pulse concentration distribution) is roughly in consistent with such concentrated distribution feature. Therefore, the simplification is used to approximate the magnitude of Marangoni force.

Figure R3 (Supplementary Figure 6 in the Supplementary Materials). Evolution of surfactant distribution. Schematics of concentration distribution with (a) and without (b) surfactant transport through diffusion and advection. (c) Numerically-calculated distribution of surfactant at $t = 0.8$ s (Supplementary Movie 7). (d) Numerically-calculated surfactant concentration along the perimeter of motor. The surfactant is concentrated around the releasing siphon. In derivation of Equation S7, we simplify the concentration around the perimeter as a pulse function denoted by the red line.

Our modification to the manuscript: In the Manuscript, we added sentences (Page 8, Lines 4-10) to explain the simplification and numerical study of surfactant concentration distribution. In the Supplementary Materials, we added Supplementary Note 3 (Page 5, Lines 17-19 and Pages 6-8) to explain details of our numerical model and Supplementary Figure 6, Supplementary Movie 7 to show results of our numerical study and verify our simplified assumption on the basis of numerical surfactant concentration distribution.

Comment 12:

(d) Unfortunately, I am not convinced with the data presented in Figs. 2(e) and (f). In 2(e), there are only 3 to 4 data points for each weight percentage, that too without any error bars.

Our response: We thank the reviewer for this insightful suggestion. By experimentally varying droplet volume V , we added more data in Figure R4 (revised Figure 2e). For each data point, experiments are repeated for 5 times so that error bar denoting the standard deviation is added.

Figure R4 (revised Figure 2e in the Manuscript). Logarithmic representation of the measured finite spreading radius R_f and its scaling prediction. Error bars denote standard deviation of 5 experiments. Colour denotes mass percentage of calcium chloride in the liquid bath. The droplet is aqueous solution of 1 wt% sodium alginate and 25 wt% PEGDA400.

Our modification to the manuscript: In the Manuscript, we added more data in Figure 2e (Page 21) and added error bar by repeating each experiment for 5 times.

Comment 13:

Similarly, in 2(f), I am not convinced that the fitting line (I am guessing which represents Eq. 3) truly represents the experimental trend.

Our response: We thank the reviewer for pointing this out. Because it is experimentally challenging to increase the order of magnitude of R_m and U_{max} in Figure 2f, therefore, the relationship indicated by the fitting line is not exceptionally convincing. We highlight the limitation of Figure 2f by stating “*Note that the scaling in Figure 2f is semi-quantitative as R_m and U_{max} span for only one order of magnitude which is the largest range that we can experimentally obtain.*” in the Manuscript.

Our modification to the manuscript: In the Manuscript, we added sentences (Page 8, Lines 16-18) to explain limitation of Figure 2f, that is, it spans for only one order of magnitude and highlight that it is only semi-quantitative because of such limitation.

Comment 14:

2. The definition of motor efficiency, and its subsequent calculation, are unclear.

Our response: We apologize for the unclear definition of motor efficiency. We added the definition of motor output $\alpha_{\max} \equiv U_{\max}/V$ as maximum speed per unit volume and the definition of efficiency $\varepsilon_{\max} \equiv mU_{\max}^2/2m_f$ as maximum kinetic energy per unit mass of surfactant. These two metrics are used for chemical motors and their sources are cited as Reference 20, 21.

20. Pena-Francesch, A., Giltinan, J. & Sitti, M. Multifunctional and biodegradable self-propelled protein motors. *Nature communications* **10**, 3188 (2019).

21. Park, J. H., Lach, S., Polev, K., Granick, S. & Grzybowski, B. A. Metal–organic framework “swimmers” with energy-efficient autonomous motility. *ACS nano* **11**, 10914-10923 (2017).

Our modification to the manuscript: In the Manuscript, we added sentences (Page 9, Lines 21 and 22) to detail the definition of motor efficiency and output and cited References 20 and 21 to provide source of such definition.

Comment 15:

3. In the conclusion, the authors' mention- “Our design works for different gelation and surfactant chemical pairs and performs exceptionally well in terms of metrics such as output and efficiency.” It is unclear to me what are the ‘design’ parameters that are controllable apart from the choice of the chemicals.

Our response: We thank the reviewer for pointing this out. We revised the inaccurate word “design” to be “approach” and highlight that the approach is based on “combination of chemicals”. In this way, the original sentence “Our design works for different gelation and surfactant chemical pairs and performs exceptionally well in terms of metrics such as output and efficiency.” is rewritten into “Our approach, the crosslinking-mediated surfactant release achieved through combination of chemicals, works for different gelation and surfactant chemical pairs and performs exceptionally well in terms of metrics such as output and efficiency.”

Our modification to the manuscript: In the Manuscript, we rewrote the original sentence “*Our design works for different gelation and surfactant chemical pairs and performs exceptionally well in terms of metrics such as output and efficiency.*” into “*Our approach, the crosslinking-mediated surfactant release achieved through combination of chemicals, works for different gelation and surfactant chemical pairs and performs exceptionally well in terms of metrics such as output and efficiency.*” (Page 12, Lines 9-12) to clarify that the main controllable parameters for the incipient approach are combination of chemicals.

Comment 16:

4. The quality of the figures, specifically some of the graphs, leaves much to be desired. To summarize, the manuscript fails to clearly explain and characterize the self-propulsion mechanism of the mantle and siphon droplet swimmer. In my opinion, the underlying physical mechanism is not rigorously explained in a quantitative manner to merit publication in Nature Communications. The applications demonstrated in the manuscript can only be useful to the broader scientific community once the underlying physical mechanism of the swimmer is fully grasped. Moreover, is it not possible for a simple camphor boat to perform these functions? Unfortunately, I cannot recommend the present manuscript for publication.

Our response: In accordance with the review’s insightful comments, we upgraded the quality of graphs, performed experiments and numerical studies so that the explanation of the physical mechanism is more rigorous. Our approach provides a strategy to extend the lifetime of Marangoni motor. Although a simple camphor boat can also, for example, actuate an interfacial sensor, it cannot achieve long timescale propulsion. It is hoped that the revised version will meet the requirements for publication.

We would like to thank the reviewer once again for expending so much time and effort in providing a review which has helped us to improve the clarity and specificity of our manuscript.

Reviewer: 3

Comment 1:

The authors present a novel form of active soft matter which uses a combination of effects to produce truly fast (10mm/s), and long lived (~100minute) activity. This is enabled the the anisotropic crosslinking of a droplet which contains a surface active agent, and a cross linking polymer. As the polymer crosslinks, the surface active agent can only be released into the liquid as a perforation is formed as a result of anisotropic crosslinking. This technique is novel, and the effect it has as a combinatorial piece of active matter is phenomenologically rich. In this manuscript, the authors explore the mechanism with PIV, characterize other material systems which perform the same effect, and characterize the energetic limits with comparisons to literature. The authors also implement this system to push a water sensor, and to operate various examples of machines. This work is significant to the field, and the implementation of an actuation-marangoni soft matter robot is novel. The work largely supports its claims, while I have some concerns about a few details about the manuscript, listed below. The work is robust, and impressive, and after adressing the following concerns, including clarifying the mechanism and language, is suitable for publication.

Our response: We sincerely thank the reviewer for the positive feedback and insightful suggestions, which have been invaluable in enhancing the manuscript. We have revised the manuscript accordingly and provided detailed point-by-point responses to each comment.

Comment 2:

Major:

- A central point of the manuscript is the relationship between the effect of the siphon, and the effect on the movement of the droplet, but this interaction is not well characterized when compared to the rest of the manuscript (which is robust). What does the perforation look like? Does the capsule perforate, or does the droplet simply not crosslink at one zone?

Our response: We appreciate the reviewer's insightful comment and apologize for the insufficient description of siphon in the original manuscript. Along three-phase contact line, the newly-crosslinked gel film is radially stretched by the Laplace pressure and surface tension force (surface tension of bath-air interface perpendicularly acting along droplet perimeter).

When local circumferential tensile stress exceeds the gel strength, crack starts to form. As shown in Figure R5, when the gel film fractures, the siphon is formed. Therefore, the perforation appears as a crack on the gel film.

Figure R5 (Supplementary Figure 3 in the Supplementary Materials). Siphon formed by gel fracture. Fluorescence image of motor (a) and magnified siphon (b) showing that the releasing hole appears as a fracture on the gel mantle. The opening has a maximum width of roughly 6% of motor's radius.

Our modification to the manuscript: In the Manuscript, we added sentences (Page 6, Lines 4 and 5) to explain that the siphon forms because of gel mantle fracture. In the Supplementary Materials, we added Supplementary Figure 3 (Page 11) to show image of the perforation.

Comment 3:

Does the "jet" of surface-active agent proceed throughout the experiment?

Our response: We thank the reviewer for pointing this out. The jet accompanies the motor motility throughout the experiment. Because approaching the end of motor's lifetime, that is, the pulsed mode, the motor only moves when jet forms (Figure R6 and Supplementary Movie 5). Without jet, the motor cannot move.

Figure R6 (Supplementary Figure 4 in the Supplementary Materials). Pulsed motion. (a) Image of a motor after 7 pulsed motions (Supplementary Movie 5). (b) Displacement, velocity, and acceleration of the 7 pulsed motions.

Our modification to the manuscript: In the Manuscript, we added sentences (Page 6, Lines 21 and 22; Page 7, Lines 1 and 2) to explain that the jet proceeds as motor propels. In the Supplementary Materials, we added Supplementary Figure 4 and Supplementary Movie 5 to show that even approaching the end of motor lifetime, that is, in the pulsed motion mode, the jet accompanies the propulsion.

Comment 4:

- Continuing in this direction, references to the importance of "laplace" pressure occur frequently in the manuscript. The curvature does increase, but at the length scale of the droplet, the authors should demonstrate that it is the Laplace pressure that is moving surfactant out of the droplet.

Our response: We apologize for insufficient discussion of Laplace-pressure-driven release in the original manuscript. As shown in Figure R7 (revised Figure 1g), the pressure inside the droplet can be well described using the interface curvature as the top of droplet is not crosslinked. Because the height of droplet is much smaller than the capillary length, thus hydrostatic pressure can be neglected. Beneath the bath-air interface, the pressure in liquid bath p_b equals to the ambient pressure p_a . By contrast, pressure inside droplet p_d is $p_a + \sigma_d \nabla \cdot \mathbf{n}$. Once the siphon forms, surfactant is pumped out because of higher p_d . The detailed discussion

is added in the Manuscript.

Figure R7 (revised Figure 1g in the Manuscript). The gradual crosslinking lowers droplet radius $R_m(t)$, increasing the Laplace pressure of liquid interior. Pressure buildup swiftly perforates the gel mantle, forming a siphon through which surfactant is released.

Our modification to the manuscript: In the Manuscript, we added sentences (Page 5, Lines 15-20) to analyse that pressure inside the droplet is higher than that outside of the droplet by an amount of $\sigma_d \nabla \cdot \mathbf{n}$. Therefore, it is the Laplace pressure which pumps surfactant out of droplet.

Comment 5:

Minor:

- The references and comparisons to the literature is somewhat incomplete, especially considering water-air interface swimmers and their introduction of anisotropy, and with reference to pulsing microrobots e.g., symmetry breaking.

Our response: We sincerely thank the reviewer for this valuable suggestion. As shown in Figure R8 (revised Figure 3f), we have classified and added water-air interface swimmer in the literature to compare with our approach. Full references of those water-air interface swimmers are listed in Supplementary Table 6.

Figure R8 (revised Figure 3f in the Manuscript). Logarithmic representation of the maximum relative velocities U/B and size B for insects, terrestrial and aquatic animals, artificial robots, and water-air interface swimmers. The light violet, blue violet, green, orange, and red shadings, respectively, denotes rigid robot, terrestrial soft robot, aquatic soft robot, water-air interface swimmer, and our gelling-droplet motor. Full references are given in Supplementary Table 6.

Our modification to the manuscript: In the Manuscript, we added data of water-air interface swimmer in Figure 3f (Page 23). In the Supplementary Materials, we updated relevant references in Supplementary Table 6 (Pages 34 and 35) to complete the comparison between our work and reported water-air interface swimmer.

Comment 6:

- The authors refer to the system as a "host-guest" system, could they elaborate on why this system qualifies as a host-guest system?

Our response: We apologize for our misuse of “host-guest system” in the original manuscript. We replaced the original “host-guest system” with “gelling system” to make the use of technical term more accurate.

Our modification to the manuscript: In the Manuscript, we revised the original “*host-guest system*” into “*gelling system*” (Page 3, Line 7; Page 8, Line 20; Page 12, Line 17) to improve description accuracy.

Comment 7:

- The authors claim that "Marangoni motor is plagued by short lifetime of ~10 s, limiting its practical application for modular powering source". It would be helpful to understand the intended object the proposed method would be used to power, and specifically what the physical limitations of traditional Marangoni-powered devices are.

Our response: We thank the reviewer for this valuable suggestion. We added that the intended object that the proposed method would be used to power, which includes devices such as sensor, drug capsule, and even mini-generator. These Marangoni-powered devices have been reported in other literature. The physical limitations of the powered devices are that they must be light-weight and floatable.

Our modification to the manuscript: In the Manuscript, we added sentences (Page 12, Lines 19 and 20) to mention the intended object to be powered and the physical limitations (floatable and light-weight) of Marangoni-powered device.

Comment 8:

- In paragraph 3 the authors state: "As shown in Figure 1c,e, upon touching solution bath, the surfactant-loading droplet spreads into a liquid lens, increasing capillary forces for its flotation⁹". What are the components of the droplet, is it more dense than water, and should I expect it to sink if not placed above the solution?

Our response: We thank the reviewer for pointing this out. The droplet is aqueous solution of sodium alginate and has a slightly higher density (1.001 g ml^{-1}) than that of water. As shown in Figure R9 and Supplementary Movie 3, in the absence of surfactant, the droplet sinks in the bath and then crosslinks into a gel sphere.

Figure R9 (images in Supplementary Movie 3). Sinking of alginate-solution droplet without surfactant.

Our modification to the manuscript: In the Manuscript, we added sentences (Page 3, Lines 10-12) to explain that without surfactant, the alginate solution droplet sinks in bath because of higher mass density. In the Supplementary Materials, we added video into Supplementary Movie 3 to show that the droplet indeed sinks without surfactant.

Comment 9:

- The order, and depiction of figure 1 should be reconsidered. The connection to the various timescales of the experiment is an interesting way to depict the various behaviors of the droplet, but is confusing. To this end, the idea that the droplet is contracting when in contact with the solution (f) is very far away from 1h, where the importance of mantle contraction becomes apparent. Furthermore, indication of the crosslinking of the droplet-water interface would be helpful.

Our response: We thank the reviewer for this valuable suggestion. In accordance with the reviewer' comment, we replotted original Figure 1f, g. As shown in Figure R10 (revised Figure 1f, g), mantle contraction is drawn in Figure 1g so that droplet contracting is close to Figure 1f. We indicated the crosslinking of the droplet-bath interface in Figure 1f. In this way, the description of Figure 1 becomes clearer.

Figure R10 (revised Figure 1f,g in the Manuscript). (f) Diffusion of calcium ions from bath to droplet triggers the chelation, forming the gel mantle which significantly cuts rapid surfactant diffusion. The crosslinking starts at droplet-bath interface and advances toward droplet bulk solution. (g) The gradual crosslinking lowers droplet radius $R_m(t)$, increasing the Laplace pressure of liquid interior. Pressure buildup swiftly perforates the gel mantle, forming a siphon through which surfactant is released.

Our modification to the manuscript: In the Manuscript, we redraw Figure 1f, g (Page 19). In the revised Figure 1f, indication of the crosslinking of the droplet-water interface is added. In the revised Figure 1g, mantle contraction is added so that the indication of such important factor is next to Figure 1f.

Comment 10:

- The authors should consider using a phrase such as self-generated, self-directed as opposed to "malice of forethought" at the end of page 3, and should take a general look at the manuscript to clarify language in and around experiment, such as the word "liberating", or "lowers" rather than "weakens" surface tension.

Our response: We thank the reviewer for this valuable suggestion. We clarified the language used in the manuscript. The phrase such as original "*malice of forethought*" was deleted; the original word "*liberating*" was replaced by "*placing*"; the original word "*weakens*" was replaced by "*lowers*". These changes improve the accuracy of language in the manuscript.

Our modification to the manuscript: In the manuscript, we deleted the original "*malice of forethought*"; we replaced the original word "*liberating*" with "*placing*" (Page 4, Line 3); we replaced the original word "*weakens*" with "*lowers*" (Page 5, Line 1). In the Supplementary

Materials, we replaced the original word “*weakens*” with “*lowers*” (Page 1, Line 11). In this way, the language used in the manuscript is clarified.

Comment 11:

- Figure 2a, the authors dyed the droplet red (alginate containing PEGDA), why is the droplet trapped in the PEGDA and not released?

Our response: We sincerely thank the reviewer for this insightful comment. We think that the reviewer intends to ask that why the red dye is trapped in the droplet and not released. We used direct red 23 to dye the droplet for visualization. Because of many free hydroxyl and carboxyl groups, the sodium alginate linear chain strongly interacts with direct red dye molecules through hydrogen bonding. In this way, red dyes are trapped inside the droplet and not released. Source of this mechanism is cited as Reference 15.

15. Chen, C., He, E., Jia, W., Xia, S. & Yu, L. Preparation of magnetic sodium alginate/sodium carboxymethylcellulose interpenetrating network gel spheres and use in superefficient adsorption of direct dyes in water. *International Journal of Biological Macromolecules* **253**, 126985 (2023).

Our modification to the manuscript: In the Manuscript, we added sentences (Page 4, Lines 6-8; Page 21, Lines 4-6) to explain why the red dye is trapped in droplet and cited Reference 15 to provide the source of the explanation.

Comment 12:

- Supplementary movie 1: the authors compare a camphor boat to their own method to demonstrate the "long-lived" nature, can the authors provide a comparison of the "energy input" into either motor? What is the relative mass of the camphor, to the propellant in the viscous example? Why does the viscous example have a stated speed of ~100s mm/s, when the authors mention ~10s of mm/s in the introduction.

Our response: We thank the reviewer for pointing this out. For both cases in Supplementary Movie 1 (left: camphor boat; right: gelling droplet, the original viscous example), same amount of surfactant is used (0.004 g PEGDA400). In both cases, boats are fabricated from a polyvinyl butyral sheet and has the same weight of 0.0029 g. The gelling-droplet powered one has extra

droplet weight of 0.016 g. For the gelling-droplet powered one, it has the maximum velocity of 141.99 mm s⁻¹ (originally stated in the supplementary movie) and an average velocity of 19.33 mm s⁻¹ (originally stated in the introduction of the Manuscript). These descriptions were added in the caption of Supplementary Figure 1.

Our modification to the manuscript: In the Supplementary Materials, we added sentences in the caption of Supplementary Figure 1 (Page 9, Lines 3-8) to provide experimental details (input surfactant mass, weight of the boat and droplet) and to explain that the gelling-droplet powered boat has the maximum velocity of ~100 mm s⁻¹ and an average velocity of ~10 mm s⁻¹.

Comment 13:

- Figure 2d/e/f: what is the wt% noted in the figure?

Our response: We apologize for insufficient description of figures. In Figure 2d, the wt% in the figure represents mass percentage of calcium chloride in the liquid bath. In Figure 2e, the wt% in the figure represents mass percentage of calcium chloride in the liquid bath. In figure 2f, wt% in the figure represents mass percentage of sodium alginate and surfactant in the droplet solution. These descriptions were added in the caption of Figure 2d, e, f.

Our modification to the manuscript: In the Manuscript, we added sentences in caption of Figure 2d, e, f (Page 21, Lines 12, 13; Page 22, Lines 1, 3, 4, 6-8) to explain the meaning of wt% in the Figure.

Comment 14:

- Can the authors clarify what surface tensions they refer to, as in to which interface is this surface tension applied?

Our response: We thank the reviewer for this valuable suggestion. Throughout the manuscript, we clarified the interface with which the interfacial tension or surface tension is associated. In this way, the readability is improved.

Our modification to the manuscript: In the Manuscript, we added words (Page 4, Line 15). In the Supplementary Materials, we added words (Page 1, Line 12; Page 3, Line 19; Page 5, Line 5) to clarify the surface tension we refer to.

Comment 15:

- The authors refer to a period of "pulsed motion", but the effect of the pulsed motion is not visible in the figure, or in any of supplementary videos. Can the authors include a note on the pulsed motion, the scale or reliability of pulsing, and a video?

Our response: We thank the reviewer for this valuable suggestion. As shown in Figure R6 (Supplementary Figure 4) and Supplementary Movie 5, we added figures and videos on the pulsed motion mode. Such pulsed motion is caused by intermittent jet containing surface-active agent.

Figure R6 (Supplementary Figure 4 in the Supplementary Materials). Pulsed motion. (a) Image of a motor after 7 pulsed motions (Supplementary Movie 5). (b) Displacement, velocity, and acceleration of the 7 pulsed motions.

Our modification to the manuscript: In the Manuscript, we added sentences (Page 6, Lines 21 and 22) to explain that the pulsed motion is associated with pulsed jet. In the Supplementary Materials, we added Supplementary Figure 4 and Supplementary Movie 5 to visualize and describe the feature of pulsed motions.

Comment 16:

- The perforated syphon should be shown in some capacity, either by microscopy, or post-experiment to demonstrate, or measure the size of the syphon.

Our response: We sincerely thank the reviewer for the insightful comment. As shown in Figure

R5 (Supplementary Figure 3), we visualize the siphon using fluorescence microscope and measure its size to be roughly 6% of motor's radius.

Figure R5 (Supplementary Figure 3 in the Supplementary Materials). Siphon formed by gel fracture. Fluorescence image of motor (a) and magnified siphon (b) showing that the releasing hole appears as a fracture on the gel mantle. The opening has a maximum width of roughly 6% of motor's radius.

Our modification to the manuscript: In the Supplementary Materials, we added Supplementary Figure 3 (Page 11) to visualize the siphon and measure its size.

Comment 17:

- Does the top of the droplet gel? Does this gelling of the top of the droplet (if it does not occur) affect the performance of the droplet?

Our response: We sincerely thank the reviewer for the insightful comment. Throughout the lifetime, motor's top remains in liquid state as the upward diffusion of calcium ion is too slow to crosslink the entire droplet. Because the droplet top is not crosslinked, it allows us to describe the pressure inside the droplet using the interface curvature. Such description was added in the Manuscript.

Our modification to the manuscript: In the Manuscript, we added sentences (Page 5, Lines 13-16) to explain that throughout the motor lifetime, the droplet top has not crosslinked.

We would like to thank the reviewer once again for expending so much time and effort in providing a review which has helped us to improve the clarity and specificity of our manuscript.

Response to Reviewer

Reviewer 2:

Comment 1:

I have now carefully gone through the authors' responses to my previous comments. I still have some questions, as mentioned below, which I think are critical to the scientific understanding of the work. I cannot recommend the publication of the present manuscript until these questions are unambiguously clarified.

Our response: We sincerely thank the reviewer for his/her valuable and constructive comments, which have greatly contributed to improving the quality of our manuscript. We have revised the manuscript accordingly and provided the following point-by-point responses to each comment.

Comment 2:

1. I found the authors' arguments regarding the perforation of the hydrogel mantle to be hand-waving at best. If I understand the system correctly then there are two types of stresses acting on an elemental area of the hydrogel mantle-- one, a compressive stress due to the local Laplace pressure (this is the stress which is dependent on the curvature), and two, a tensile stress due to the inherent tendency of the interfacial tension to minimize the surface area. If the perforation of the hydrogel mantle is dependent on the curvature (i.e. the shell perforates only once a critical shape is reached), then the compressive stress due to the Laplace pressure plays a dominant role in the rupture of the mantle. This is only my conjecture; the authors must explain it better. I think that even if the exact location of the rupture is not deterministic in nature, the mechanism of the rupture should be clearly explained. Otherwise, it will be impossible to engineer the system.

Our response: We thank the reviewer for the insightful comment. We numerically mapped the distribution of stress in the mantle before perforation so that the mechanism can be clearly explained. As shown in Figure R1, because of negligible hydrostatic force, the mantle is subjected to surface tension force σ_{da} , σ_{ba} acting on its perimeter and Laplace pressure $\sigma_{da} \nabla \cdot \mathbf{n}$ acting on its interior surface. As the curvature increases, the Laplace pressure and angle θ_d , that

is, the direction of surface tension force σ_{da} , increase. We calculated the distribution of von Mises stress (T_{von}) in the mantle subjected to these dominant external forces using finite element analysis. It is found that the maximum von Mises stress appears at the perimeter of the mantle. The position of the maximum von Mises stress corresponds to the region where perforation preferentially occurs.

Figure R1 (Supplementary Figure 3 in the Supplementary Materials). Stress in mantle before perforation. (a) Because of negligible hydrostatic force, the mantle is subjected to surface tension force σ_{da} , σ_{ba} and Laplace pressure $\sigma_{da} \nabla \cdot \mathbf{n}$. (b) Schematics of configuration for finite element analysis. Surface tension forces, $\sigma_{da} = 42 \text{ mN m}^{-1}$ and $\sigma_{ba} = 70 \text{ mN m}^{-1}$, act along the contact line which is denoted blue. Laplace pressure, $\sigma_{da} \nabla \cdot \mathbf{n} = 12.24 \text{ N m}^{-2}$, acts on inner surface of the mantle. Based on experiment, $\theta_d = 30^\circ$ and $\theta_b = 2^\circ$. Centre of contact line ring is set as origin. B(0, 0, -1 mm) is the bottom point of the mantle. R_m is set to be 2.8 mm. For boundary conditions, the contact line remains on water surface, thus its displacement in z -direction is 0. Because of axisymmetric configuration, the displacement of point B in x - and y -direction is 0. Properties of the calcium alginate gel are set as follows: Young's modulus is 30 kPa, Poisson's ratio is 0.5, and density is 2200 kg m^{-3} . (c) The numerically-calculated von Mises stress (T_{von}) in the mantle. The maximum stress appears at the perimeter where perforation preferentially occurs.

Our modification to the manuscript: In the Manuscript, we added sentences (Page 6, Line4) to explain that stress peaks at mantle's perimeter where perforation preferentially occurs. In the

Supplementary Materials, we added Supplementary Figure 3 to show the numerical stress distribution in the mantle.

Comment 3:

2. Supplementary Figure 5 (Figure R1): What does the constant value of the radius R_m represent, especially for the micromotors in calcium chloride solution? If gelation continues, then I would expect R_m to be a function of time. In fact, the authors also consider the same during the evaluation of the liquid release rate from the micromotor (Eq. S3). Then what does the constant value of R_m represent in Figure S5/R1 (b) for a given concentration of calcium chloride?

If $R_m(t)$ variation is insignificant as the authors claim, then the variation in the average curvature may also be insignificant. If that is the case, then why should the liquid release rate vary over time? More importantly, why should the speed U decay over time? Is $R(t)$ variation really negligible even for $t \sim O(10^2-10^3)$ s? Note that Figure 2(d) shows the variation only for the first 3 seconds. How would this variation look for $t \sim O(10^3)$ s?

I would request the authors to show the decay of U over time in water, as well as in calcium chloride solutions of different concentrations.

Our response: We sincerely thank the reviewer for the constructive comment. We apologize for the insufficient description of R_m in Supplementary Figure 6b (original Supplementary Figure 5b), the R_m here is the micromotor's radius measured at the end of its lifetime.

We update original Figure 2d by providing a longer-timescale (~ 1000 s) evolution of micromotor's radius (Figure R2), the motor's radius R is indeed a function of time as the crosslinking continues throughout its propulsion. The motor's radius rapidly decreases in the very first ~ 50 s and then gradually decreases at a much lower rate in the rest ~ 1000 s.

As shown in Figure R3, we provide the decay of U over time in water (30 s in CaCl_2 solution and then in water) as well as in calcium chloride solutions (maintained in CaCl_2 solution) of different concentrations (0.5 and 2 wt%). Velocities decay in an exponential manner.

The propelling velocity U relates to the surfactant releasing rate J through

$U \sim \sqrt[3]{\frac{\kappa J}{\rho_b C_d \pi R_m h_m}}$ (Equation S9). In physical rather than phenomenological perspective, flow

of surfactant solution through the siphon is approximated as the Poiseuille flow driven by the

Laplace pressure and has an expression $J \sim c_m Q \sim 10^{-3} c_m \frac{R_m(t)^3 \sigma_d \nabla \cdot \mathbf{n}(t)}{\mu_d}$ (see Equation S3).

The motor radius R_m as well as the droplet-air curvature $\nabla \cdot \mathbf{n}$ are both time-dependent. The droplet curvature is impacted by two factors, that is, the volume of droplet (Figure R4) and the motor radius R_m which sets the boundary of droplet. Although R_m largely plateaus after the very first ~ 50 s, the continuous decrease in droplet volume decreases the curvature and thus brings about decay in velocity.

Figure R2 (Updated Figure 2d in the Manuscript). Temporal evolution of the droplet radius $R(t)/R_0$ for a longer timescale (~ 1000 s). The moment when droplet touches the bath is defined as $t = 0$. Colour denotes mass percentage of calcium chloride in the liquid bath. The droplet is aqueous solution of 1 wt% sodium alginate and 25 wt% PEGDA400.

Figure R3 (Supplementary Figure 6e,f in the Supplementary Materials). The normalized propelling velocity as a function of time for 0.5-wt% (a) and 2-wt% (b) CaCl_2 solution.

Figure R4. Schematics of droplets in different volume. Droplet-air curvature is impacted by droplet's volume.

Our modification to the manuscript: In the Manuscript, we updated Figure 2d so that a longer-timescale (~ 1000 s) evolution of micromotor's radius is provided. In the Supplementary Materials, we added sentences (Page 14, Lines 7,8) to explain that the R_m in Supplementary Figure 6b is the micromotor's radius measured at the end of lifetime and sentences (Page 3, Lines 2-4) to explain that the droplet curvature is impacted by two factors, that is, the volume of droplet and the motor radius; we added Supplementary Figure 6e,f to show the decay of U over time in water (30 s in CaCl_2 solution and then in water) as well as in calcium chloride solution (maintained in CaCl_2 solution) of different concentrations (0.5 and 2 wt%)

Comment 4:

3. Figure 2b (Figure R2): At present, the form of the exponential fit to the decay of U holds no physical justification. I would request the authors to investigate whether the coefficient and the characteristic decay time can be estimated (at least the order of magnitude) from the explanation for the propulsion dynamics presented in section 2 of the supplementary material. Currently, 53 and 0.0016^{-1} have no physical meaning. Moreover, the right-hand side Y-axis should be L/D .

Our response: We sincerely thank the reviewer for this valuable and constructive comment. We estimate the first-order releasing rate constant and characteristic decay time from the explanation of the propulsion dynamics in Supplementary Note 2. By substituting equation S5 into S9, we have the relationship between temporal evolution of propulsion velocity and the

first-order release of surfactant as $U(t) \sim \left(\frac{\kappa c_m Q_0}{\rho_b C_d \pi R_m h_m} \right)^{1/3} e^{-kt/3}$. By referring to the fitted

exponential decay of velocity $U/D = 53e^{-0.0016t}$ in Figure 2b (propelling velocity after initial

5 s is exponentially fitted), we estimate the releasing rate constant by $k/3 = 0.0016 \text{ s}^{-1}$, giving $k = 0.0048 \text{ s}^{-1}$ and a characteristic decay time $1/k \sim 208 \text{ s}$. The coefficient $\left(\frac{\kappa c_m Q_0}{\rho_b C_d \pi R_m h_m} \right)^{1/3}$ with a unit of m s^{-1} is the maximum propulsion velocity which is associated with the coefficient 53 BL s^{-1} in the exponential fitting of propulsion velocity. We have changed the right-hand side Y-axis to be L/D .

Our modification to the manuscript: In the Manuscript, we added sentences (Page 8, Lines 17-19) to explain the derivation of temporal propulsion velocity and changed the right-hand side Y-axis of Figure 2b to be L/D . In the Supplementary Materials, we added sentences (Page 5, Lines 17,18 and Page 6, Lines 1-3) to detail the derivation of temporal propulsion velocity and estimate the releasing rate constant k and characteristic decay time of the first-order release.

Comment 5:

4. I am still confused with Fig. 2e. Equation S2 does not have a prefactor; therefore, in the log-log plot there should be no intercept. Right now for the plot to have slope 1, there has to be an intercept. Is there a physical genesis for the intercept?

Our response: We thank the reviewer for giving us the opportunity to clarify the interception in Figure 2e. Figure 2e is the logarithmic representation of the measured finite spreading radius

R_1 and its scaling prediction $\left(\frac{\Delta\sigma V^3}{\pi^3 \rho_d D_s^2} \right)^{1/8}$. The slope of 1 is captured but the prediction

overestimates measured R_1 by roughly an order of magnitude so that an intercept appears. The intercept is potentially due to simplification in the assumption of spreading cessation and estimation of diffusion coefficient.

For the assumption of spreading cessation, when diffusion homogenizes the surfactant at the leading edge, the driving force, that is, surface tension difference $\Delta\sigma$ vanishes so that the spreading stops. Similar to Kim *et al.* (*Nature Physics* **13**, 1105-1110), it is assumed that such scenario happens when the surfactant diffusion boundary layer thickness l_d equals to the lens thickness h . This is a physically-motivated but simplified criterion, particularly for the quantitative relationship between l_d and h . For example, the spreading may stop when $h = bl_d$

which gives $R_1 \sim b^{-1/2} \left(\frac{\Delta\sigma V^3}{\pi^3 \rho_d D_s^2} \right)^{1/8}$. When $b = 4$, the prediction decreases by a factor of 0.5.

For the estimation of diffusion coefficient D_s , it is estimated by the Stokes-Einstein model as $D_s = k_b T / (6\pi\mu_b r)$. On the basis of previous works on polyethylene glycol (*Biophysical journal* **95**, 1590–1599; *Membranes* **8**, 23), we assume the hydrodynamic radius r of our surfactant poly(ethylene glycol) diacrylate 400 to be 0.5 nm, giving a diffusion coefficient of $\sim 10^{-10} \text{ m}^2 \text{ s}^{-1}$. The Stokes-Einstein model is most accurate for spherical particle in a continuous solvent where the particle is much larger than the solvent molecules. However, diffusion of surfactant violates the ideal prerequisite as polymer chain is not rigid sphere, size of polymer is not significantly small than that of water molecules, boundary condition for polymer chain may not be no-slip, and interaction between polymer chains happens. These factors contribute to deviation in the estimation of diffusion coefficient. For example, the actual diffusion

coefficient may be cD_s which gives $R_1 \sim c^{-1/4} \left(\frac{\Delta\sigma V^3}{\pi^3 \rho_d D_s^2} \right)^{1/8}$. When $c = 16$, the prediction decreases by a factor of 0.5.

By analysing the simplification in derivation of Equation S2, we propose that the intercept in Figure 2e is potentially due to simplification in the assumption of spreading cessation and estimation of diffusion coefficient.

Our modification to the manuscript: To explain that the intercept is potentially due to simplification in the assumption of spreading cessation and estimation of diffusion coefficient, in the Manuscript, we added sentences (Page 22, Lines 5-7); in the Supplementary Materials, we added sentences (Page 2, Lines 11-14).

Comment 6:

5. The major concern with Fig. 2f is that the spread in the data is too great to reach any meaningful conclusion. In fact, on apparent inspection, a line with a higher slope seems to be a better fit than the line with slope $1/3$.

Our response: We thank the reviewer for pointing this out. In previous Figure 2f, data from droplets of different compositions are plotted, causing a wide spread. To obtain a reliable

relationship, we perform experiments using only one composition (aqueous solution of 0.375 wt% sodium alginate and 25 wt% PEGDA400). Droplets' sizes are controlled through different nozzles. For a specific type of nozzle, experiments are repeated 5 times so that error bars denoting standard deviation are provided. As shown in Figure R5, the data are distributed much closer to a fitting line whose slope is roughly 1/3.

Figure R5 (updated Figure 2f in the Manuscript). The maximum propulsion velocity U_{\max} as a function of the motor radius R_m . The red line is the fitting line. Error bars denote standard deviation of 5 experiments. The droplet is aqueous solution of 0.375 wt% sodium alginate and 25 wt% PEGDA400. The liquid bath is 1 wt% calcium chloride aqueous solution.

Our modification to the manuscript: In the Manuscript, we updated Figure 2f through only one composition (aqueous solution of 0.375 wt% sodium alginate and 25 wt% PEGDA400) and repeated experiments.

Comment 7:

6. The definition of efficiency is still unclear. I did not understand what the authors mean by 'per unit mass of surfactant'. Can the authors please explain how they calculate this quantity?

Our response: We thank the reviewer for giving us the opportunity to clarify the definition of efficiency. Unlike conventional definition of efficiency, that is, work output divided by energy input, the efficiency used in our manuscript is firstly defined by Steve Granick and Bartosz A.

Grzybowski (*ACS Nano* **11**, 10914–10923). In their work, they define it as “*the particle’s maximum kinetic energy per unit mass of expended fuel*”. The particle’s maximum kinetic energy is calculated based on the maximum propulsion velocity as $mU_{\max}^2/2$ where m is the initial total mass of motor which includes mass of motor matrix and loaded fuel. The total mass of fuel (surfactant) m_f loaded in their metal-organic framework motor is measured using the thermogravimetric analysis. As a result, the efficiency is calculated as $\epsilon_{\max} \equiv mU_{\max}^2/2m_f$. In our work, we calculate the efficiency by following their definition. For example, for a motor of initial total mass $m = 0.01$ g wherein mass of surfactant is $m_f = 0.0025$ g, we measure its maximum velocity to be 890 mm s^{-1} , then its efficiency is $mU_{\max}^2/2m_f = 1584.2 \text{ } \mu\text{J g}^{-1}$.

Our modification to the manuscript: In the Manuscript, we added sentences (Page 10, Lines 4,5) to explain the definition of variables in the equation of efficiency.

We would like to thank the reviewer once again for expending so much time and effort in providing a review which has helped us to improve the clarity and specificity of our manuscript.

Response to Reviewer

Reviewer 2:

Comment:

I have gone through the authors' replies to my previous round of comments. I am satisfied with their replies. I think the manuscript has achieved the quality and rigour necessary for publication in Nature Communications. Hence, I recommend the manuscript for publication.

Our response: We sincerely thank the reviewer for his/her recommendation for publication of our manuscript. We thank the reviewer for expending so much time and effort in providing a review which has helped us to improve the clarity and specificity of our manuscript.